# Pre-training Contextualized World Models with In-the-wild Videos for Reinforcement Learning

**Jialong Wu,**[*] **Haoyu Ma,**[*] **Chaoyi Deng, Mingsheng Long**[⊠]
School of Software, BNRist, Tsinghua University, China
`wujialong0229@gmail.com, {mhy22,dengcy23}@mails.tsinghua.edu.cn`
`mingsheng@tsinghua.edu.cn`

## Abstract

Unsupervised pre-training methods utilizing large and diverse datasets have achieved tremendous success across a range of domains. Recent work has investigated such unsupervised pre-training methods for model-based reinforcement learning (MBRL) but is limited to domain-specific or simulated data. In this paper, we study the problem of pre-training world models with abundant in-the-wild videos for efficient learning of downstream visual control tasks. However, in-the-wild videos are complicated with various contextual factors, such as intricate backgrounds and textured appearance, which precludes a world model from extracting shared world knowledge to generalize better. To tackle this issue, we introduce Contextualized World Models (ContextWM) that explicitly separate context and dynamics modeling to overcome the complexity and diversity of in-the-wild videos and facilitate knowledge transfer between distinct scenes. Specifically, a contextualized extension of the latent dynamics model is elaborately realized by incorporating a context encoder to retain contextual information and empower the image decoder, which encourages the latent dynamics model to concentrate on essential temporal variations. Our experiments show that in-the-wild video pre-training equipped with ContextWM can significantly improve the sample efficiency of MBRL in various domains, including robotic manipulation, locomotion, and autonomous driving. Code is available at this repository: `https://github.com/thuml/ContextWM`.

## 1 Introduction

Model-based reinforcement learning (MBRL) holds the promise of sample-efficient learning for visual control. Typically, a world model [20, 40] is learned to approximate state transitions and control signals of the environment to generate imaginary trajectories for planning [22] or behavior learning [71]. In the wake of revolutionary advances in deep learning, world models have been realized as action-conditional video prediction models [34] or latent dynamics models [22, 21]. However, given the expressiveness and complexity of deep neural networks, the sample efficiency of MBRL can still be limited by the failure to learn an accurate and generalizable world model efficiently.

*Pre-training and fine-tuning* paradigm has been highly successful in computer vision [82, 26] and natural language processing [57, 13] to fast adapt pre-trained representations for downstream tasks, while learning *tabula rasa* is still dominant in MBRL. Recent work has taken the first step towards pre-training a world model, named Action-free Pre-training from Videos (APV) [68]. However, it has been conducted by pre-training on domain-specific and carefully simulated video datasets, rather than abundant *in-the-wild* video data [17, 18, 30, 83]. Prior attempts of leveraging real-world video data result in either underfitting for video pre-training or negligible benefits for downstream visual control tasks [68]. Against this backdrop, we naturally ask the following question:

*Can world models pre-trained on diverse in-the-wild videos benefit sample-efficient learning of downstream visual control tasks?*

---

[*]Equal Contribution

37th Conference on Neural Information Processing Systems (NeurIPS 2023).

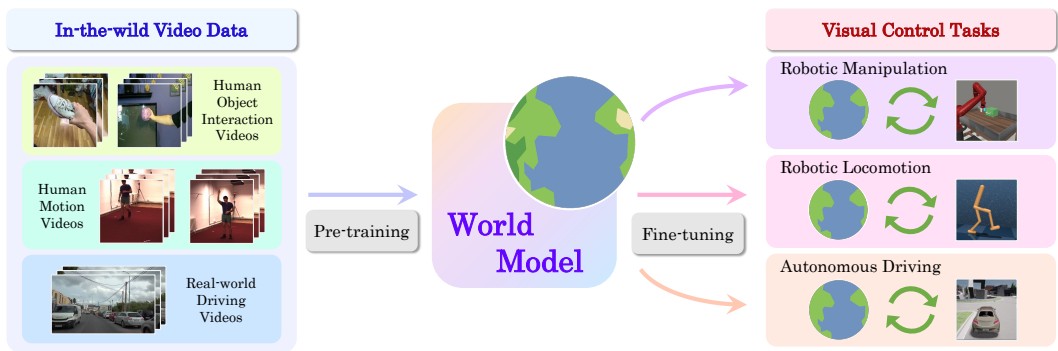

Figure 1: Illustration of In-the-wild Pre-training from Videos (IPV). In various domains, we pre-train world models with in-the-wild videos by action-free video prediction (*left*) and then fine-tune the pre-trained one on downstream visual control tasks with model-based reinforcement learning (*right*).

It opens up a way to utilize the vast amount of videos available on the Internet and thereby fully release the potential of this methodology by applying it at scale. Large and diverse data spanning various scenes and tasks can provide world knowledge that is widely generalizable and applicable to a variety of downstream tasks. For instance, as depicted in Figure 1, world models for robotic manipulation tasks can probably benefit not only from videos of humans interacting with target objects in diverse ways but also from motion videos that embody rigorous physical rules. Analogous to APV [68], we brand our paradigm as In-the-wild Pre-training from Videos (IPV).

However, in-the-wild videos are of inherent high complexity with various factors, such as intricate backgrounds, appearance, and shapes, as well as complicated dynamics. Coarse and entangled modeling of context and dynamics can waste a significant amount of model capacity on modeling low-level visual details of *what is there* and prevent world models from capturing essential shared knowledge of *what is happening*. Biological studies reveal that in natural visual systems, $\sim$80% retinal ganglion cells are P-type operating on spatial detail and color, while $\sim$20% are M-type operating on temporal changes [28, 44]. Partially inspired by this, we suggest that leveraging IPV for world models calls for not only appropriate data and model scaling [35] but more essentially, dedicated design for context and dynamics modeling.

In this paper, we present the Contextualized World Model (ContextWM), a latent dynamics model with visual observations that explicitly separates context and dynamics modeling to facilitate knowledge transfer of semantically similar dynamics between visually diverse scenes. Concretely, a contextualized latent dynamics model is derived to learn with variational lower bound [33, 38] and elaborately realized by incorporating a context encoder to retain contextual information and a parsimonious latent dynamics model to concentrate on essential temporal variations. Moreover, dual reward predictors are introduced to enable simultaneously enhancing task-relevant representation learning as well as learning an exploratory behavior. The main contributions of this work are three-fold:

- From a data-centric view, we systematically study the paradigm of pre-training world models with in-the-wild videos for sample-efficient learning of downstream tasks, which is much more accessible and thus makes it possible to build a general-purpose world model.
- We propose Contextualized World Models (ContextWM), which explicitly model both the context and dynamics to handle complicated in-the-wild videos during pre-training and also encourage task-relevant representation learning when fine-tuning with MBRL.
- Our experiments show that equipped with our ContextWM, in-the-wild video pre-training can significantly improve the sample efficiency of MBRL on various domains.

## 2 Related Work

**Pre-training in RL.** Three categories of pre-training exist in RL: unsupervised online pre-training, offline pre-training, and visual representation pre-training. The first two are both domain-specific, which learn initial behaviors, primitive skills, or domain-specific representations by either online environment interaction [55, 39] or offline demonstrations [41, 3]. Recent work has explored general-

purpose visual pre-training using in-the-wild datasets [69, 36, 77, 53, 86, 49, 45] to accelerate policy learning in model-free manners. Plan2Explore [66], as a model-based method, pre-trains its world model and exploratory policy simultaneously by online interaction. Ye *et al.* [81] and Xu *et al.* [78] build upon EfficientZero [80] and pre-train their world models using offline experience datasets across Atari games. To the best of our knowledge, APV [68] is the first to pre-train world models with datasets from different domains, though the datasets are manually simulated demonstrations by scripted policies across tasks from RLBench [32], which still lacks diversity and scale.

**Visual control with in-the-wild videos.** While several existing works leverage demonstration videos for learning visual control [43, 84, 62], rare efforts have been made to utilize off-the-shelf video datasets from the Internet. Shao *et al.* [70] and Chen *et al.* [9] learn reward functions using videos from the Something-Something dataset [17]. R3M [49] leverages diverse Ego4D dataset [18] to learn reusable visual representations by time contrastive learning. Other applications with such videos include video games [7, 6], visual navigation [8], and autonomous driving [88]. We instead pre-train world models by video prediction on in-the-wild video datasets.

**World models for visual RL.** World models that explicitly model the state transitions and reward signals are widely utilized to boost sample efficiency in visual RL. A straightforward method is to learn an action-conditioned video prediction model [50, 34] to generate imaginary trajectories. Ha and Schmidhuber [20] propose to first use a variational autoencoder to compress visual observation into latent vectors and then use a recurrent neural network to predict transition on compressed representations. Dreamer [22, 21, 23], a series of such latent dynamics models learning via image reconstruction, have demonstrated their effectiveness for both video games and visual robot control. There also have been other methods that learn latent representations by contrastive learning [51, 11] or value prediction [63, 25]. Recent works also use Transformers [75] as visual encoders [67] or dynamics models [10, 47, 58] that make world models much more scalable and can be complementary with our effort to pre-train world models with a vast amount of diverse video data.

Concurrent to our work, SWIM [46] also pre-trains a world model from large-scale human videos and then transfers it to robotic manipulation tasks in the real world. However, it designs a structured action space via visual affordances to connect human videos with robot tasks, incorporating a strong inductive bias and thus remarkably limiting the range of pre-training videos and downstream tasks.

## 3  Background

**Problem formulation.** We formulate a visual control task as a partially observable Markov decision process (POMDP), defined as a tuple $(\mathcal{O}, \mathcal{A}, p, r, \gamma)$. $\mathcal{O}$ is the observation space, $\mathcal{A}$ is the action space, $p(o_t \mid o_{<t}, a_{<t})$ is the transition dynamics, $r(o_{\leq t}, a_{<t})$ is the reward function, and $\gamma$ is the discount factor. The goal of MBRL is to learn an agent $\pi$ that maximizes the expected cumulative rewards $\mathbb{E}_{p,\pi}[\sum_{t=1}^{T} \gamma^{t-1} r_t]$, with a learned world model $(\hat{p}, \hat{r})$ approximating the unknown environment. In our pre-training and fine-tuning paradigm, we also have access to an in-the-wild video dataset $\mathcal{D} = \{(o_t)_{t=1}^{T}\}$ without actions and rewards to pre-train the world model.

**Dreamer.** Dreamer [21, 23, 24] is a visual model-based RL method where the world model is formulated as a latent dynamics model [16, 22] with the following four components:

$$
\begin{aligned}
&\text{Representation model:} \quad z_t \sim q_\theta(z_t \mid z_{t-1}, a_{t-1}, o_t) \qquad &&\text{Image decoder:} \quad \hat{o}_t \sim p_\theta(\hat{o}_t \mid z_t) \\
&\text{Transition model:} \quad \hat{z}_t \sim p_\theta(\hat{z}_t \mid z_{t-1}, a_{t-1}) \qquad &&\text{Reward predictor:} \quad \hat{r}_t \sim p_\theta(\hat{r}_t \mid z_t)
\end{aligned} \tag{1}
$$

The representation model, also known as *posterior* of $z_t$, approximates latent state $z_t$ from previous state $z_{t-1}$, previous action $a_{t-1}$ and current observation $o_t$, while the transition model, also known as *prior* of $z_t$, predicts it directly from $z_{t-1}$ and $a_{t-1}$. The overall models are jointly learned by minimizing the negative variational lower bound (ELBO) [33, 38]:

$$
\mathcal{L}(\theta) \doteq \mathbb{E}_{q_\theta(z_{1:T} \mid a_{1:T}, o_{1:T})} \Big[ \sum_{t=1}^{T} \Big( -\ln p_\theta(o_t \mid z_t) - \ln p_\theta(r_t \mid z_t) \tag{2}
$$
$$
+ \beta_z \, \mathrm{KL}\left[ q_\theta(z_t \mid z_{t-1}, a_{t-1}, o_t) \,\|\, p_\theta(\hat{z}_t \mid z_{t-1}, a_{t-1}) \right] \Big) \Big].
$$

Behavior learning of Dreamer can be conducted by actor-critic learning purely on imaginary latent trajectories, for which we refer readers to Hafner *et al.* [21] for details.

**Action-free Pre-training from Videos (APV).** To leverage action-free video data, APV [68] first pre-trains an action-free variant of the latent dynamics model (with representation $q_\theta(z_t \mid z_{t-1}, o_t)$, transition $p_\theta(\hat{z}_t \mid z_{t-1})$ and image decoder $p_\theta(\hat{o}_t \mid z_t)$) as a video prediction model, which drops action conditions and the reward predictor from Eq. (1) and Eq. (2). When fine-tuned for MBRL, APV stacks an action-conditional model (with representation $q_\phi(s_t \mid s_{t-1}, a_{t-1}, z_t)$ and transition $p_\phi(\hat{s}_t \mid s_{t-1}, a_{t-1})$) on top of the action-free one. Action-free dynamics models and image decoders are initialized with pre-trained weights $\theta$ from video pre-training. Furthermore, to utilize pre-trained representations for better exploration, APV introduces a video-based intrinsic bonus $r_t^{\texttt{int}}$ measuring the diversity of visited trajectories by random projection and nearest neighbors. The overall models during fine-tuning are optimized by minimizing the following objective:

$$\mathcal{L}(\phi, \theta) \doteq \mathbb{E}_{q_\phi(s_{1:T}|a_{1:T}, z_{1:T}), q_\theta(z_{1:T} \mid o_{1:T})} \Big[ \sum_{t=1}^{T} \Big( \underbrace{-\ln p_\theta(o_t|s_t)}_{\text{image log loss}} \underbrace{-\ln p_\phi(r_t + \lambda r_t^{\texttt{int}}|s_t)}_{\text{reward log loss}} \tag{3}$$

$$\underbrace{+\beta_z \, \mathrm{KL}\left[q_\theta(z_t|z_{t-1}, o_t) \,\|\, p_\theta(\hat{z}_t|z_{t-1})\right]}_{\text{action-free KL loss}} \quad \underbrace{+\beta_s \, \mathrm{KL}\left[q_\phi(s_t|s_{t-1}, a_{t-1}, z_t) \,\|\, p_\phi(\hat{s}_t|s_{t-1}, a_{t-1})\right]}_{\text{action-conditional KL loss}} \Big) \Big].$$

# 4 Contextualized World Models

In this section, we present Contextualized World Models (ContextWM), a framework for both action-free video prediction and visual model-based RL with explicit modeling of context and dynamics from observations. Our method introduces: (i) a contextualized extension of the latent dynamics model (see Section 4.1), (ii) a concrete implementation of contextualized latent dynamics models tailored for visual control (see Section 4.2). We provide the overview and pseudo-code of our ContextWM in Figure 3 and Appendix A, respectively.

## 4.1 Contextualized Latent Dynamics Models

While latent dynamics models have been successfully applied for simple and synthetic scenes, in-the-wild images and videos are naturalistic and complex. Our intuition is that there are two groups of information in the sequence of observations, namely time-invariant *context* and time-dependent *dynamics* [76, 12, 42, 15]. The context encodes static information about objects and concepts in the scene, such as texture, shapes, and colors, while the dynamics encode the temporal transitions of the concepts, such as positions, layouts, and motions. To overcome the complexity of in-the-wild videos, it is necessary to explicitly represent the complicated contextual information, thus shared knowledge with respect to dynamics can be learned and generalized across distinct scenes.

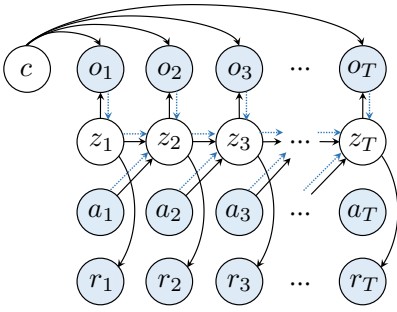

Figure 2: Probabilistic graphical model of a contextualized latent dynamics model, assuming time-invariant context $c$ and time-dependent dynamics $z_t$. Solid lines show the generative model and dotted lines show variational inference.

We thus propose a contextualized extension of latent dynamics models [16, 22], extending Eq. (1). As shown in Figure 2, the probabilistic model generates the observation $o_t$ conditioned not only on the current latent dynamics $z_t$ but also on a context variable $c$ that can include rich information (solid lines in Figure 2), while the variational inference model approximates the posterior of the latent dynamics $z_t$ conditioned on observations $o_t$ and previous states $z_{t-1}$, *without* referring to the context (dotted lines in Figure 2). The proposed design allows contextualized image decoders to reconstruct diverse and complex observations using rich contextual information, expanding their ability beyond the expressiveness of latent dynamics variables $z_t$. On the other hand, through the variational information bottleneck [5, 1], latent dynamics variables $z_t$ are encouraged to selectively exclude contextual information and only capture essential temporal variations that are not included in the context $c$. Furthermore, the design of *context-unaware* latent dynamics inference encourages representation and transition models to learn this temporal information within a high-level semantic space, in contrast to merely differentiating with the context frame in the low-level visual space, facilitating the acquisition of representations more transferable and robust to intricate contexts.

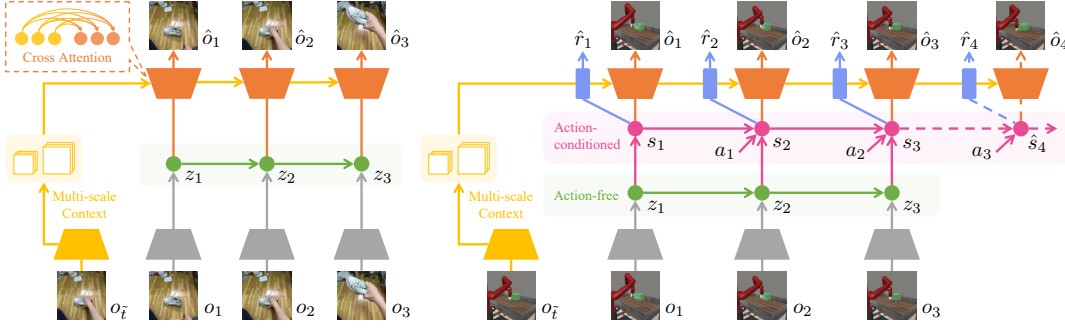

|  |  |
|---|---|
| (a) Action-free video pre-training | (b) Action-conditioned fine-tuning with MBRL |

Figure 3: Architecture of Contextualized World Models. Building upon the stacked latent model from APV [68], we empower the **image decoder** by incorporating a **context encoder** that operates in parallel with the **latent dynamics model**. The context encoder captures rich contextual information from a randomly selected context frame and enhances the decoder features of each timestep with a multi-scale cross-attention mechanism that enables flexible information shortcuts across spatial positions. This design encourages the latent dynamics model to focus only on essential temporal variations, while also allowing the decoder to reconstruct complex observations more effectively.

As derived in Appendix B, the overall model can be learned with ELBO of conditional log probability $\ln p_\theta(o_{1:T}, r_{1:T} \mid a_{1:T}, c)$, without the need to model the complex distribution of contexts $p(c)$:

$$\mathcal{L}(\theta) \doteq \underbrace{\mathbb{E}_{q_\theta(z_{1:T}\mid a_{1:T}, o_{1:T})}}_{\text{context-unaware latent inference}} \left[ \sum_{t=1}^{T} \left( \underbrace{-\ln p_\theta(o_t \mid z_t, c)}_{\text{contextualized image loss}} - \ln p_\theta(r_t \mid z_t) \right. \right. \tag{4}$$
$$\left. \left. + \beta_z \, \mathrm{KL} \left[ q_\theta(z_t \mid z_{t-1}, a_{t-1}, o_t) \, \| \, p_\theta(\hat{z}_t \mid z_{t-1}, a_{t-1}) \right] \right) \right].$$

## 4.2 Contextualized World Model Architectures

We then introduce a concrete implementation of contextualized latent dynamics models, specifically tailored for visual control. Our approach builds upon Dreamer [21], which utilizes a Recurrent State Space Model (RSSM) [22]. To enable action-free pre-training from videos and action-conditioned fine-tuning for MBRL, we also incorporate the stacked latent model from APV [68] into our design. Notably, our work distinguishes itself from previous research through the elaborate design of contextualized components and a dual reward predictor structure that enhances task-relevant representation learning. An overview of the overall architecture is illustrated in Figure 3.

**Context formulation.** There exist various choices of contextual information to be conditioned on, including text descriptions, pre-trained representations, semantic maps, or more sophisticated structured data [59]. As our focus is on end-to-end visual control, we choose the simplest one solely based on visual observations: we consider the context $c$ as a single frame of observation randomly sampled from the trajectory segment $o_{1:T}$: $c \doteq o_{\tilde{t}}$, $\tilde{t} \sim \mathrm{Uniform}\{1, 2, \dots, T\}$. It is assumed that contextual information lies equally in each frame, and by randomly selecting a context frame, our context encoder should learn to be robust to temporal variations.

**Multi-scale cross-attention conditioning.** Given the context in the form of a frame of the same size as the reconstruction $\hat{o}_t$, a U-Net [60] architecture would be one of the most appropriate choices to empower a contextualized image decoder $p_\theta(\hat{o}_t \mid z_t, c)$, as its multi-scale shortcuts help to propagate the information directly from context to reconstruction. Moreover, in our case, the image decoder learns by also conditioning on latent dynamics $z_t$, as shown in Figure 3. While a conventional U-Net architecture incorporates shortcut features directly by concatenation or summation and thus forces a spatial alignment between them, temporal variations such as motions or deformations cannot be neglected in our case. Inspired by recent advances in generative models [59, 4], we augment the decoder feature $X \in \mathbb{R}^{c \times h \times w}$ with the context feature $Z \in \mathbb{R}^{c \times h \times w}$ by a cross-attention mechanism

[75], shown as follows (we refer to Appendix C.3 for details):

$$\text{Flatten: } Q = \text{Reshape}\,(X) \in \mathbb{R}^{hw \times c}, \; K = V = \text{Reshape}\,(Z) \in \mathbb{R}^{hw \times c}$$
$$\text{Cross-Attention: } R = \text{Attention}\left(QW^Q, KW^K, VW^V\right) \in \mathbb{R}^{hw \times c} \tag{5}$$
$$\text{Residual-Connection: } X = \text{ReLU}\left(X + \text{BatchNorm}\left(\text{Reshape}\,(R)\right)\right) \in \mathbb{R}^{c \times h \times w}.$$

**Dual reward predictors.** For simplicity of behavior learning, we only feed the latent variable $s_t$ into the actor $\pi(a_t|s_t)$ and critic $v(s_t)$, thus introducing no extra computational costs when determining actions. However, this raises the question of whether the latent variable $s_t$ contains sufficient task-relevant information for visual control and avoids taking shortcuts through the U-Net skip connections. For example, in robotics manipulation tasks, the positions of static target objects may not be captured by the time-dependent $s_t$, but by the time-invariant context $c$. We remark that reward predictors $p_\phi(r_t|s_t)$ play a crucial role in compelling the latent variable $s_t$ to encode task-relevant information, as the reward itself defines the task to be accomplished. Nevertheless, for hard-exploration tasks such as Meta-world [85], the video-based intrinsic bonus [68] may distort the exact reward regressed for behavioral learning and constantly drift during the training process [61], making it difficult for the latent dynamics model to capture task-relevant signals. Therefore, we propose a dual reward predictor structure comprising a *behavioral* reward predictor that regresses the exploratory reward $r_t + \lambda r_t^{\mathrm{int}}$ for behavior learning, and additionally, a *representative* reward predictor that regresses the pure reward $r_t$ to enhance task-relevant representation learning [31].

**Overall objective.** The model parameters of ContextWM can be jointly optimized as follows:

$$\mathcal{L}^{\mathtt{CWM}}(\phi, \varphi, \theta) \doteq \underbrace{\mathbb{E}_{q_\phi(s_{1:T}|a_{1:T}, z_{1:T}), q_\theta(z_{1:T}\,|\,o_{1:T})}}_{\text{context-unaware latent inference}} \Bigg[ \sum_{t=1}^{T} \Big( \underbrace{-\ln p_\theta(o_t|s_t, c)}_{\text{contextualized image loss}}$$
$$\underbrace{-\ln p_\phi(r_t + \lambda r_t^{\mathtt{int}}|s_t)}_{\text{behavioral reward loss}} \underbrace{-\beta_r \ln p_\varphi(r_t|s_t)}_{\text{representative reward loss}} \underbrace{+\beta_z\, \mathrm{KL}\left[q_\theta(z_t|z_{t-1}, o_t)\,\|\,p_\theta(\hat{z}_t|z_{t-1})\right]}_{\text{action-free KL loss}} \tag{6}$$
$$\underbrace{+\beta_s\, \mathrm{KL}\left[q_\phi(s_t|s_{t-1}, a_{t-1}, z_t)\,\|\,p_\phi(\hat{s}_t|s_{t-1}, a_{t-1})\right]}_{\text{action-conditional KL loss}} \Big) \Bigg].$$

When pre-trained from in-the-wild videos, ContextWM can be optimized by minimizing an objective that drops action-conditioned dynamics and reward predictors from Eq. (6), similar to APV [68]. For behavior learning, we adopt the actor-critic learning scheme of DreamerV2 [23]. For a complete description of pre-training and fine-tuning of ContextWM for MBRL, we refer to Appendix A.

## 5  Experiments

We conduct our experiments on various domains to evaluate In-the-wild Pre-training from Videos (IPV) with Contextualized World Models (ContextWM), in contrast to plain world models (WM) used by DreamerV2 and APV [2]. Our experiments investigate the following questions:

- Can IPV improve the sample efficiency of MBRL?
- How does ContextWM compare to a plain WM quantitatively and qualitatively?
- What is the contribution of each of the proposed techniques in ContextWM?
- How do videos from different domains or of different amounts affect IPV with ContextWM?

### 5.1  Experimental Setup

**Visual control tasks.** As shown in Figure 4, we formulate our experiments on various visual control domains. Meta-world [85] is a benchmark of 50 distinct robotic manipulation tasks and we use the same six tasks as APV [68]. DMC Remastered [19] is a challenging extension of the widely used robotic locomotion benchmark, DeepMind Control Suite [73], by expanding a complicated graphical variety. We also conduct an autonomous driving task using the CARLA simulator [14], where an agent needs to drive as far as possible along Town04's highway without collision in 1000 timesteps. See Appendix C.2 for details on visual control tasks.

---

[2]In our experiments, we rebrand the APV baseline [68] with the new name 'IPV w/ Plain WM' to note that it is pre-trained on **I**n-the-wild data rather than only **A**ction-free data.

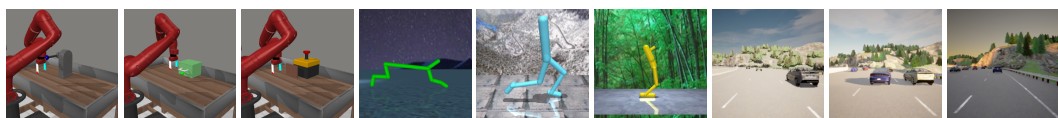

Figure 4: Example image observations of our visual control benchmark tasks: Meta-world, DMControl Remastered, and CARLA (*left to right*).

**Pre-training datasets.** We utilize multiple in-the-wild video datasets which can potentially benefit visual control. Something-Something-v2 (SSv2) [17] dataset contains 193K videos of people interacting with objects. Human3.6M [30] dataset contains videos of human poses over 3M frames under 4 different viewpoints. YouTube Driving [88] dataset collects 134 real-world driving videos, over 120 hours, with various weather conditions and regions. We also merge the three datasets to construct an assembled dataset for the purpose of pre-training a general world model. See Figure 1 for example video frames and Appendix C.1 for details on data preprocessing.

**Implementation details.** For the newly introduced hyperparameter, we use $\beta_r = 1.0$ in Eq. (6) for all tasks. To ensure a sufficient capacity for both plain WM and ContextWM pre-trained on diverse in-the-wild videos, we implement visual encoders and decoders as 13-layer ResNets [27]. Unless otherwise specified, we use the same hyperparameters with APV. See Appendix C.3 for more details.

**Evaluation protocols.** Following Agarwal *et al.* [2] and APV, we conduct 8 individual runs for each task and report interquartile mean with bootstrap confidence interval (CI) for individual tasks and with stratified bootstrap CI for aggregate results.

## 5.2 Meta-world Experiments

**SSv2 pre-trained results.** Figure 5a shows the learning curves on six robotic manipulation tasks from Meta-world. We observe that in-the-wild pre-training on videos from SSv2 dataset consistently improves the sample efficiency and final performance upon the DreamerV2 baseline. Moreover, our proposed ContextWM surpasses its plain counterpart in terms of sample efficiency on five of the six tasks. Notably, the dial-turn task proves challenging, as neither method is able to solve it. These results demonstrate that our method of separating context and dynamics modeling during pre-training and fine-tuning can facilitate knowledge transfer from in-the-wild videos to downstream tasks.

**Ablation study.** We first investigate the contribution of pre-training in our framework and report the aggregate performance with and without pre-training at the top of Figure 5b. Our results show that a plain WM seldom benefits from pre-training, indicating that the performance gain of IPV with plain WM is primarily due to the intrinsic exploration bonus. This supports our motivation that the complex contexts of in-the-wild videos can hinder knowledge transfer. In contrast, ContextWM significantly improves its performance with the aid of video pre-training. Additionally, we evaluate the contribution of the proposed techniques in ContextWM, as shown at the bottom of Figure 5b. We experiment with replacing the cross-attention mechanism (Eq. (5)) with simple concatenation or removing the dual reward predictor structure. Our results demonstrate that all these techniques contribute to the performance of ContextWM, as both variants outperform the plain WM.

**Effects of dataset size.** To investigate the effects of pre-training dataset size, we subsample 1.5k and 15k videos from SSv2 dataset to pre-train ContextWM. Figure 7a illustrates the performance of ContextWM with varying pre-training dataset sizes. We find that pre-training with only a small subset of in-the-wild videos can almost match the performance of pre-training with the full data. This is probably because, despite the diversity of contexts, SSv2 dataset still lacks diversity in dynamics patterns as there are only 174 classes of human-object interaction scenarios, which can be learned by proper models with only a small amount of data. It would be interesting for future work to explore whether there is a favorable scaling property w.r.t. pre-training dataset size when pre-training with more sophisticated video datasets with our in-the-wild video pre-training paradigm.

**Effects of dataset domain.** In Figure 7b, we assess pre-training on various video datasets, including RLBench [32] videos curated by APV [68] and our assembled data of three in-the-wild datasets. We observe that while pre-training always helps, pre-training from a more similar domain, RLBench, outperforms that from SSv2. Nevertheless, simulated videos lack diversity and scale and make it difficult to learn world models that are broadly applicable. Fortunately, in-the-wild video pre-training

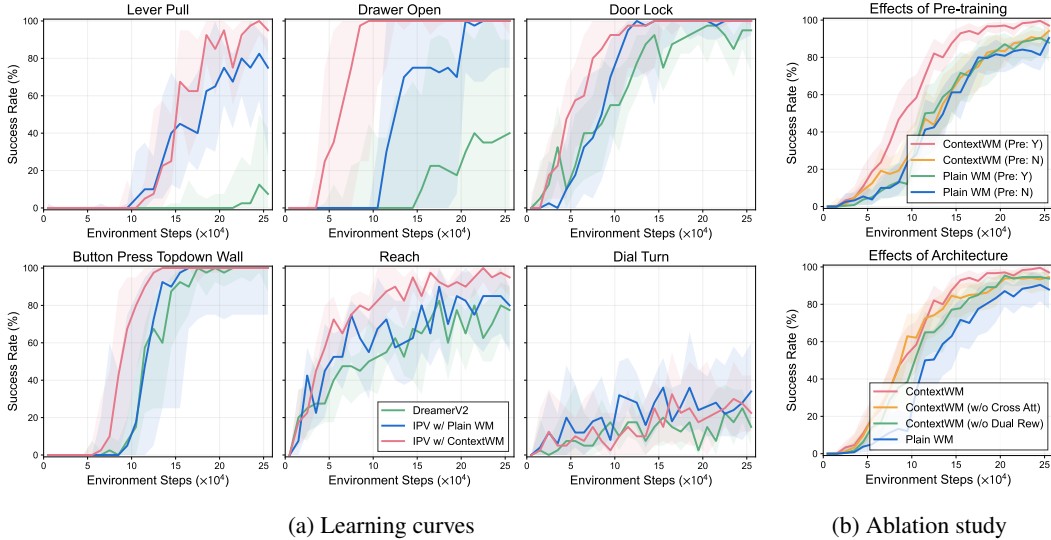

(a) Learning curves

(b) Ablation study

Figure 5: Meta-world results. (a) Learning curves with in-the-wild pre-training on SSv2 dataset, as measured on the success rate, aggregated across eight runs. (b) Performance of ContextWM and plain WM with or without pre-training (*top*) and performance of ContextWM and its variants that replace the cross-attention mechanism with naive concatenation or remove the dual reward predictor structure (*bottom*). We report aggregate results across a total of 48 runs over six tasks.

can continuously improve its performance with a more diverse assembled dataset, suggesting a promising scalable alternative for domain-specific pre-training.

### 5.3 DMControl Remastered Experiments

In the top row of Figure 6a, we present the learning curves of IPV from SSv2 dataset and DreamerV2 on the DMC Remastered locomotion tasks. Remarkably, we observe that pre-training with the SSv2 dataset can significantly enhance the performance of ContextWM, even with a large domain gap between pre-training and fine-tuning. This finding suggests that ContextWM effectively transfers shared knowledge of separately modeling context and dynamics. In contrast, a plain WM also benefits from pre-training, but it still struggles to solve certain tasks, such as hopper stand. These results also suggest that our ContextWM could be a valuable practice for situations where visual generalization is critical, as ContextWM trained from scratch also presents a competitive performance in Figure 6b.

**Effects of dataset domain.** Figure 7b demonstrates the performance of pre-training on human motion videos from the Human3.6M dataset. However, we observe a negative transfer, as pre-training from Human3.6M leads to inferior performance compared to training from scratch. We argue that the Human3.6M dataset is collected in the laboratory environment, rather than truly *in-the-wild*. These results support our motivation that pre-training data lacking in diversity can hardly help learn world models that are generally beneficial. Additionally, we experiment with pre-training on the assembled data of three video datasets but find no significant improvement over SSv2 pre-training.

### 5.4 CARLA Experiments

The bottom row of Figure 6a displays the learning curves of IPV from the SSv2 dataset and DreamerV2 on the CARLA driving task under different weather and sunlight conditions. Similar to the DMC Remastered tasks, we find that MBRL benefits from pre-training on the SSv2 dataset, despite a significant domain gap. We observe that in almost all weather conditions, IPV with ContextWM learns faster than plain WM at the early stages of training and can also outperform plain WM in terms of the final performance. Minor superiority of ContextWM over plain WM also suggests that in autonomous driving scenarios, a single frame may not contain sufficient contextual information, and more sophisticated formulations of contextual information may further enhance performance.

**Effects of dataset domain.** We then assess the performance of ContextWM on CARLA by pre-training it from alternative domains. As shown in Figure 7b, we observe that pre-training from

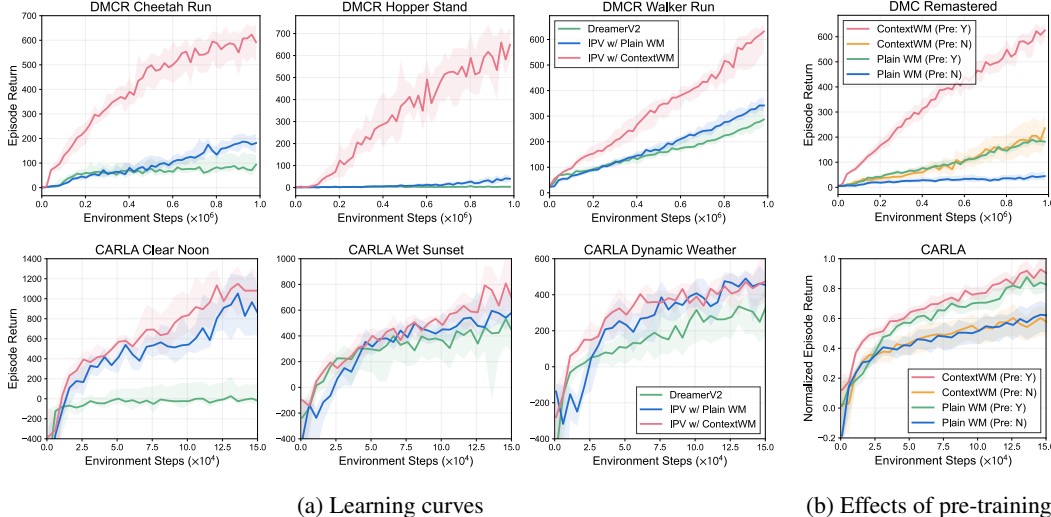

(a) Learning curves                                               (b) Effects of pre-training

Figure 6: DMC Remastered (*top*) and CARLA (*bottom*) results. (a) Learning curves with pre-training on SSv2 dataset, as measured on the episode return, aggregated across eight runs. (b) Performance of ContextWM and plain WM with or without pre-training, aggregated across 24 runs over three tasks. Episode returns of each task in CARLA are normalized to comparable ranges.

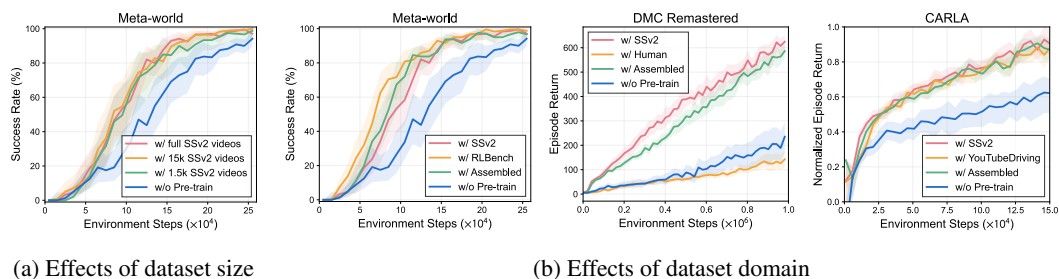

(a) Effects of dataset size                                    (b) Effects of dataset domain

Figure 7: Analysis on pre-training datasets. We report aggregated results over tasks. (a) Performance of ContextWM on Meta-world tasks, with varying pre-training dataset sizes. (b) Performance of ContextWM with pre-training on videos from various domains.

the YouTube Driving dataset or a combination of three in-the-wild datasets can both improve upon learning from scratch. However, neither can significantly surpass the performance achieved by pre-training from SSv2, despite a narrower domain gap between YouTube Driving and CARLA. We conjecture that this could be attributed to the higher complexity of the YouTube Driving dataset in comparison to SSv2. It would be promising to explore the potential of scaling ContextWM further to capture valuable knowledge from more complex videos.

## 5.5 Qualitative Analysis

**Video prediction.** We visually investigate the future frames predicted by a plain WM and ContextWM on SSv2 in Figure 8a. We find that our model effectively captures the shape and motion of the object, while the plain one fails. Moreover, we also observe that cross-attentions in our model (Eq. (5)) successfully attend to varying spatial positions of the context to facilitate the reconstruction of visual details. This shows our model works with better modeling of context and dynamics.

**Video representations.** We sample video clips of length 25 with two distinct labels (*push something from right to left* and *from left to right*) and visualize the averaged model states of the sampled videos using t-SNE [74] in Figure 8b. Note that we do not utilize any labels of the videos in pre-training. Video representations of plain WM may entangle with extra contextual information and thus are not sufficiently discriminative to object motions. However, ContextWM which separately models context and dynamics can provide representations well distributed according to different directions of motion.

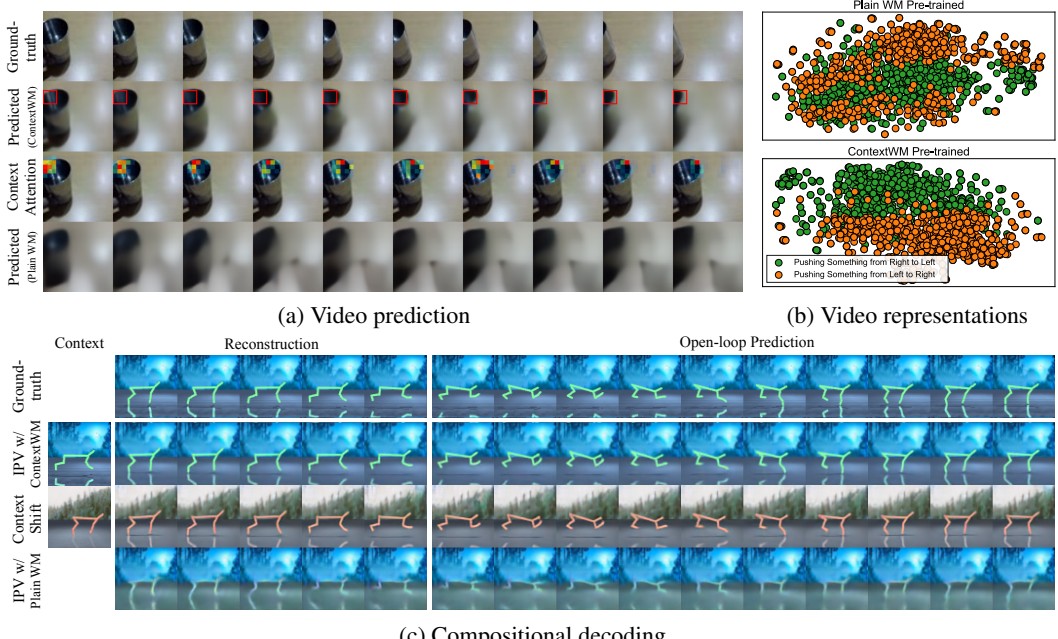

(a) Video prediction

(b) Video representations

(c) Compositional decoding

Figure 8: Qualitative analysis. (a) Future frames predicted on SSv2. Predictions from our model well capture the shape and motion of the water cup. Moreover, our context cross-attentions from the left upper corner (red box) of different frames successfully attend to varying spatial positions of the context frame. (b) t-SNE visualization of average pooled representations of SSv2 videos with two distinct labels, from a pre-trained plain WM and ContextWM, respectively. (c) Compositional decoding analysis on the DMCR domain, where fine-tuned ContextWM is able to combine the new context with dynamics information from the original trajectory.

**Compisitional decoding.** We conduct a compositional decoding analysis by sampling a random frame from another trajectory to replace the original context and leaving dynamics the same. As shown in Figure 8c, ContextWM correctly combines the new context with the original dynamics information. These results show that our model, fine-tuned on the DMCR domain, has successfully learned disentangled representations of contexts and dynamics. In contrast, the plain WM suffers from learning entangled representations and thus makes poor predictions about the transitions.

## 6 Discussion

This paper presents Contextualized World Models (ContextWM), a framework for both action-free video prediction and visual model-based RL. We apply ContextWM to the paradigm of In-the-wild Pre-training from Videos (IPV), followed by fine-tuning on downstream tasks to boost learning efficiency. Experiments demonstrate the effectiveness of our method in solving a variety of visual control tasks from Meta-world, DMC Remastered, and CARLA. Our work highlights not only the benefits of leveraging abundant in-the-wild videos but also the importance of innovative world model design that facilitates knowledge transfer and scalable learning.

**Limitations and future work.** One limitation of our current method is that a randomly selected single frame may not sufficiently capture complete contextual information of scenes in real-world applications, such as autonomous driving. Consequently, selecting and incorporating multiple context frames as well as multimodal information [59] for better context modeling need further investigation. Our work is also limited by medium-scale sizes in terms of both world models and pre-training data, which hinders learning more broadly applicable knowledge. An important direction is to systematically examine the scalability of our method by leveraging scalable architectures like Transformers [47, 67] and massive-scale video datasets [18, 48]. Lastly, our work focuses on pre-training world models via generative objectives, which utilize the model capacity inefficiently on image reconstruction to overcome intricate contexts. Exploring alternative pre-training objectives, such as contrastive learning [51, 11] or self-prediction [65, 80], could further release the potential of IPV by eliminating heavy components on context modeling and focusing on dynamics modeling.

## Acknowledgments

We would like to thank many colleagues, in particular Haixu Wu, Baixu Chen, and Jincheng Zhong, for their valuable discussion. This work was supported by the National Key Research and Development Plan (2020AAA0109201), the National Natural Science Foundation of China (62022050 and 62021002), and the Beijing Nova Program (Z201100006820041).

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

## A Pseudocode

For clarity, we first elaborate on the optimization objective for contextualized world models during action-free pre-training, which drops action-conditioned dynamics and reward predictors from the fine-tuning objective (Eq. (6)):

$$\mathcal{L}^{\text{CWM-pt}}(\theta) \doteq \underbrace{\mathbb{E}_{q_\theta(z_{1:T} \mid o_{1:T})}}_{\text{context-unaware latent inference}} \left[ \sum_{t=1}^{T} \left( \underbrace{-\ln p_\theta(o_t|z_t, c)}_{\text{contextualized image loss}} \right. \right. \tag{7}$$

$$\left. \left. \underbrace{+\beta_z \, \text{KL} \left[ q_\theta(z_t|z_{t-1}, o_t) \,\|\, p_\theta(\hat{z}_t|z_{t-1}) \right]}_{\text{action-free KL loss}} \right) \right].$$

For behavior learning, we adopt the same actor-critic learning scheme based purely on imaginary latent trajectories $\hat{s}_\tau, \hat{a}_\tau, \hat{r}_\tau$ with horizon $H$, as DreamerV2 [21, 23]. The critic $v_\xi(s)$ is learned by regressing the $\lambda$-target [72, 64]:

$$\mathcal{L}^{\text{critic}}(\xi) \doteq \mathbb{E}_{p_\phi, \pi_\psi} \left[ \sum_{\tau=t}^{t+H} \frac{1}{2} \left( v_\xi(\hat{s}_\tau) - \text{sg}(V_\tau^\lambda) \right)^2 \right], \tag{8}$$

$$V_\tau^\lambda \doteq \hat{r}_\tau + \gamma \begin{cases} (1-\lambda)v_\xi(\hat{s}_{\tau+1}) + \lambda V_{\tau+1}^\lambda & \text{if } \tau < t + H \\ v_\xi(\hat{s}_{\tau+1}) & \text{if } \tau = t + H, \end{cases} \tag{9}$$

where sg is a stop gradient function. And the actor $\pi_\psi(a|s)$ is learned by maximizing the imagined return by back-propagating the gradients through the learned world model, with an entropy regularizer:

$$\mathcal{L}^{\text{actor}}(\psi) \doteq \mathbb{E}_{p_\phi, \pi_\psi} \left[ \sum_{\tau=t}^{t+H} \left( -V_\tau^\lambda - \eta \, \text{H} \left[ \pi_\psi(a_\tau|\hat{s}_\tau) \right] \right) \right], \tag{10}$$

Overall, a complete description of pre-training and fine-tuning of ContextWM for MBRL is presented in Algorithm 1.

## B Derivations

The variational bound for contextualized latent dynamics models $p(o_{1:T}, r_{1:T} \mid a_{1:T}, c)$ and a variational posterior $q(z_{1:T} \mid a_{1:T}, o_{1:T})$ follows from importance weighting and Jensen's Inequality as shown [3],

$$\ln p(o, r|a, c) \geq \mathbb{E}_{q(z|a,o)} \left[ \ln \frac{p(o, r, z|a, c)}{q(z|a, o)} \right]$$

$$= \mathbb{E}_{q(z|a,o)} \left[ \ln p(o, r|z, a, c) \right] - \text{KL} \left[ q(z|a, o) \| p(z|a) \right]$$

$$= \mathbb{E}_{q(z|a,o)} \left[ \sum_{t=1}^{T} \ln p(o_t|z_t, c) + \ln p(r_t|z_t) \right] - \text{KL} \left[ q(z|a, o) \| p(z|a) \right],$$

where

$$\text{KL} \left[ q(z|a, o) \| p(z|a) \right] = \int_z q(z|a, o) \ln \prod_{t=1}^{T} \frac{q(z_t|z_{t-1}, a_{t-1}, o_t)}{p(z_t|z_{t-1}, a_{t-1})} \mathrm{d}z$$

$$= \sum_{t=1}^{T} \left[ \int_{z_{1:t-1}} \mathrm{d}z_{1:t-1} \, q(z_{1:t-1}|o, a) \int_{z_t} \mathrm{d}z_t \, q(z_t|z_{t-1}, a_{t-1}, o_t) \ln \frac{q(z_t|z_{t-1}, a_{t-1}, o_t)}{p(z_t|z_{t-1}, a_{t-1})} \right.$$

$$\left. \int_{z_{t+1:T}} \mathrm{d}z_{t+1:T} \, q(z_{t+1:T}|a, o) \right]$$

$$= \sum_{t=1}^{T} \mathbb{E}_{q(z_{t-1}|a,o)} \text{KL} \left[ q(z_t|z_{t-1}, a_{t-1}, o_t) \| p(z_t|z_{t-1}, a_{t-1}) \right].$$

---

[3]For brevity, we denote $z_{1:T}, a_{1:T}, o_{1:T}, r_{1:T}$ as $z, a, o, r$, respectively.

---

**Algorithm 1** Contextualized World Models with In-the-wild Pre-training from Videos

---

1: **// Action-free pre-training with in-the-wild videos**
2: Initialize parameters $\theta$ of action-free dynamics model, image encoder, and decoder randomly
3: Load in-the-wild video dataset $\mathcal{D}$
4: **for** each training step **do**
5:      // WORLD MODEL PRE-TRAINING
6:      Sample random minibatch $\{o_{1:T}\} \sim \mathcal{D}$
7:      Get context $c \leftarrow o_{\tilde{t}}, \; \tilde{t} \sim \text{Uniform}\{1, T\}$
8:      Update world model by minimizing $\mathcal{L}^{\texttt{CWM-pt}}(\theta)$ in Eq. (7)
9: **end for**

10: **// Action-conditioned fine-tuning with MBRL**
11: Load pre-trained parameters $\theta$ of action-free dynamics model, image encoder, and decoder
12: Initialize parameters $\phi, \varphi$ of action-conditioned dynamics model and reward predictors randomly
13: Initialize parameters $\psi, \xi$ of actor $\pi_\psi(a|s)$ and critic $v_\xi(s)$
14: Initialize replay buffer $\mathcal{B}$ with random seed episodes
15: **for** each timestep $t$ **do**
16:     // COLLECT TRANSITIONS
17:     Get representation $z_t \sim q_\theta(z_t|z_{t-1}, a_{t-1}, o_t), s_t \sim q_\phi(s_t|s_{t-1}, a_{t-1}, z_t)$
18:     Get action $a_t \sim \pi_\psi(a_t|s_t)$
19:     Add transition $\{(o_t, a_t, r_t)\}$ to replay buffer $\mathcal{B}$

20:     // WORLD MODEL FINE-TUNING
21:     Sample random minibatch $\{(o_{1:T}, a_{1:T}, r_{1:T})\} \sim \mathcal{B}$
22:     Get context $c \leftarrow o_{\tilde{t}}, \; \tilde{t} \sim \text{Uniform}\{1, T\}$
23:     Update world model by minimizing $\mathcal{L}^{\texttt{CWM}}(\phi, \varphi, \theta)$ in Eq. (6)

24:     // BEHAVIOR LEARNING
25:     Imagine future rollouts $\{\hat{s}_\tau, \hat{a}_\tau, \hat{r}_\tau + \lambda \hat{r}_\tau^{\texttt{int}}\}$ using world model and actor
26:     Update actor and critic by minimizing objectives in Eq. (8) and Eq. (10)
27: **end for**

---

Overall, we have a lower bound on the log-likelihood of the data:

$$\ln p(o, r|a, c) \geq \mathbb{E}_{q(z|a,o)} \left[ \sum_{t=1}^{T} \left( \ln p(o_t|z_t, c) + \ln p(r_t|z_t) - \text{KL}\left[q(z_t|z_{t-1}, a_{t-1}, o_t) \| p(z_t|z_{t-1}, a_{t-1})\right] \right) \right].$$

We can also calculate that the tightness of the bound is $\text{KL}\left[q(z|a, o) \| p(z|a, o, r, c)\right]$.

## C    Experimental Details

### C.1    Data Preprocessing

We use three common video datasets for in-the-wild video pre-training. All inputs are resized to $64 \times 64$ pixels.

**Something-Something-v2.** The Something-Something-v2 dataset[4] [17] shows footage of humans interacting with common everyday objects. We use all videos in the training set but filter out videos less than 25 frames long, resulting in a total of 162K videos for pre-training.

**Human3.6M.** The Human3.6M dataset[5] [30] contains videos of various activities performed by 11 human subjects. We use the particular dataset processed by Pavlakos *et al.* [56], which includes a total of 840 videos with 210 scenarios and 4 different viewpoints for each scenario. Each video has around 500 frames. To make sure the images used for pre-training better fit our downstream tasks, we place the subject at the center of each frame by cutting a bounding box calculated from the 3D joint positions provided by the dataset. In addition, a padding of 200 pixels is performed around

---

[4]`https://developer.qualcomm.com/software/ai-datasets/something-something`
[5]`http://vision.imar.ro/human3.6m/description.php`

the bounding box so that the subject is properly positioned. The image clipped using the padded bounding box is then resized to 64×64 pixels.

**YouTube Driving.**   The YouTube Driving dataset[6] [88] consists of real-world driving videos with different road and weather conditions. As videos in this dataset usually have a short opening, the first 100 frames of each video are skipped. The original video is downsampled 5 times. We employ all 134 videos in our pre-training process, with each video containing roughly 10000 to 20000 frames.

**Assembled Dataset.**   We compose the three datasets mentioned above into an assembled dataset. On each sample, we uniformly choose a dataset from the three choices and sample videos from that dataset accordingly.

## C.2   Benchmark Environments

**Meta-world.**   Meta-world [85] is a benchmark of 50 distinct robotic manipulation tasks. Following APV [68], we use a subset of six tasks, namely Lever Pull, Drawer Open, Door Lock, Button Press Topdown Wall, Reach, and Dial Turn. In all tasks, the episode length is 500 steps without any action repeat. The action dimension is 4, and the reward ranges from 0 to 10. All methods are trained over 250K environment steps, which is consistent with the setting of APV [68].

**DMC Remastered.**   The DMC Remastered (DMCR) Suite [19] is a variant of the DeepMind Control Suite [73] with randomly generated graphics emphasizing visual diversity. On initialization of each episode for both training and evaluation, the environment makes a random sample for 7 factors affecting visual conditions, including floor texture, background, robot body color, target color, reflectance, camera position, and lighting. Our agents are trained and evaluated on three tasks: Cheetah Run, Hopper Stand, and Walker Run. All variation factors are used except camera position in the Hopper Stand task, where we find it too difficult for the agent to learn when the camera is randomly positioned and rotated. Following the common setup of DeepMind Control Suite [21, 79], we set the episode length to 1000 steps with an action repeat of 2, and the reward ranges from 0 to 1. All methods are trained over 1M environment steps.

**CARLA.**   CARLA [14] is an autonomous driving simulator. We conduct a task using a similar setting as Zhang *et al.* [87], where the agent's goal is to progress along a highway in 1000 time-steps and avoid collision with 20 other vehicles moving along. The maximum episode length is set to 1000 without action repeat. Possible reasons to end an episode early include deviation from the planned route, driving off the road, and being stuck in traffic due to collisions. Each action in the CARLA environment is made up of a steering dimension and an accelerating dimension (or braking for negative acceleration). In addition to the reward function in Zhang *et al.*, we add a lane-keeping reward to prevent our agent from driving on the shoulder. Our reward function is as follows:

$$r_t = v_{\text{ego}}^T \hat{u}_h \cdot \Delta t \cdot |\text{center}| - \lambda_1 \cdot \text{impluse} - \lambda_2 \cdot |\text{steer}|$$

The first term represents the effective distance traveled on the highway, where $v_{\text{ego}}$ is the velocity vector projected onto the highway's unit vector $\hat{u}_h$, multiplied by a discretized time-step $\Delta t = 0.05$ and a lane-keeping term $|\text{center}|$ to penalize driving on the shoulder. The $\text{impulse}$ term represents collisions during driving, while the $\text{steer} \in [-1, 1]$ term prevents excessive steering. The hyperparameters $\lambda_1$ and $\lambda_2$ are set to $10^{-3}$ and 1, respectively. The central camera of the agent, observed as $64 \times 64$ pixels, is used as our observation. We use three different weather and sunlight conditions offered by CARLA for training and evaluation: ClearNoon, WetSunset, and dynamic weather. In the dynamic weather setting, a random weather and sunlight condition is sampled at the start of each episode and changes realistically during the episode. To aggregate per-weather scores, we normalize raw episode returns of each weather to comparable ranges. Namely, we linearly rescale episode returns from $[-400, 1200]$ under ClearNoon, $[-400, 800]$ under WetSunset, and $[-400, 600]$ under dynamic weather to a unified range of $[0, 1]$. All methods are trained over 150K environment steps.

---

[6]`https://github.com/metadriverse/ACO`

| stage name | output size | image encoder |
|:---:|:---:|:---:|
| conv_in | 32×32 | 3×3, 48, stride 2 |
| stage1 | 16×16 | $\begin{bmatrix} 3\times3,\ 48 \\ 3\times3,\ 48 \end{bmatrix}\times2$ |
| | | average pool 2×2, stride 2 |
| stage2 | 8×8 | $\begin{bmatrix} 3\times3,\ 96 \\ 3\times3,\ 96 \end{bmatrix}\times2$ |
| | | average pool 2×2, stride 2 |
| stage3 | 4×4 | $\begin{bmatrix} 3\times3,\ 192 \\ 3\times3,\ 192 \end{bmatrix}\times2$ |
| | | average pool 2×2, stride 2 |

Table 1: Architecture for image encoders in world models. Building residual blocks [27] are shown in brackets, with the number of blocks stacked. Numbers inside each bracket indicate the kernel size and output channel number of the corresponding building block, e.g. 3×3, 192 means one of the convolutions in the block uses a 3×3 kernel and the number of output channels is 192. We use Batch Normalization [29] and ReLU activation function.

## C.3  Model Details

**Overall model.**  We adopt the stack latent model introduced by APV [68] and extend it into a Contextualized World Model (ContextWM) with the following components:

> **Action-free**
> $\begin{cases}$ Representation model: $\qquad z_t \sim q_\theta(z_t \,|\, z_{t-1}, o_t)$
> Transition model: $\qquad\quad \hat{z}_t \sim p_\theta(\hat{z}_t \,|\, z_{t-1})$
>
> **Action-conditional**
> $\begin{cases}$ Representation model: $\qquad s_t \sim q_\phi(s_t \,|\, s_{t-1}, a_{t-1}, z_t)$
> Transition model: $\qquad\quad \hat{s}_t \sim p_\phi(\hat{s}_t \,|\, s_{t-1}, a_{t-1})$
>
> **Dual reward predictors**
> $\begin{cases}$ Representative reward predictor: $\qquad \hat{r}_t \sim p_\varphi(\hat{r}_t \,|\, s_t)$
> Behavioral reward predictor: $\qquad \hat{r}_t + \lambda\hat{r}_t^{\text{int}} \sim p_\phi(\hat{r}_t + \lambda\hat{r}_t^{\text{int}} \,|\, s_t)$
>
> **Contextualized image decoder**: $\qquad \hat{o}_t \sim p_\theta(\hat{o}_t \,|\, s_t, c).$ $\qquad\qquad$ (11)

Specifically, the action-free representation model utilizes a ResNet-style *image encoder* to process observation input $o_t$. Moreover, the contextualized image decoder also incorporates a similar *context encoder* to extract contextual information from the context frame $c$ and augment the decoder features with a cross-attention mechanism. We elaborate on the details below.

**Image encoders and decoder.**  To ensure a sufficient capacity for both plain WM and ContextWM pre-trained on diverse in-the-wild videos, we implement image encoders and decoders as 13-layer ResNets [27]. For the sake of simplicity, we only show the architecture of the image encoder in Table 1, as the architecture of the context encoder is exactly the same, and the architecture of the image decoder is symmetric to the encoders. In the decoder, we use nearest-neighbor upsampling for unpooling layers. The outputs of the last residual block of two stages in the context encoder (stage2 and stage3) before average pooling (thus in the shape of $16 \times 16$ and $8 \times 8$ respectively) are passed to the corresponding residual block of the image decoder and used to augment the incoming decoder features with cross-attention.

**Context cross-attention.**  The input features $X \in \mathbb{R}^{c \times h \times w}$ of residual blocks in the image decoder are augmented with shortcut features $Z \in \mathbb{R}^{c \times h \times w}$ of the same shape from the context encoder, by a cross-attention mechanism as Eq. (5). Here we elaborate on the details. After $Z$ is flattened into a sequence of tokens $\in \mathbb{R}^{hw \times c}$, to mitigate the quadratic complexity of cross-attention, we random

mask 75% of the tokens, resulting tokens of the length $\lfloor \frac{1}{4} hw \rfloor$. Moreover, we use learned position embedding $\in \mathbb{R}^{hw \times c}$ initialized with zeros for both $Q$ and $K, V$, respectively.

**Context augmentation.** We randomly sample a frame of observation from the trajectory segment as the input to the context encoder. To prevent the image decoder from taking a shortcut by trivially copying the context frame and thus impeding the training process, we apply random Cutout [89] as image augmentation on the context frame for all Meta-world tasks (except drawer open), as Meta-World observations have a static background and a fixed camera position. Note that we do not utilize any augmentation for DMCR and CARLA tasks. Our comparison with plain WM is fair since we *do not* introduce any augmentation into the image encoder of the latent dynamics models, thus the agent determines its actions solely based on the original observation.

**Latent dynamics model.** The stacked latent model from APV [68] is adopted to enable action-free pre-training and action-conditioned fine-tuning, where both the action-free and action-conditioned models are built upon the discrete latent dynamics model introduced in DreamerV2 [23], where the latent state consists of a deterministic part and a discrete stochastic part. Inputs to the latent dynamics model are representations from the image encoder. Following APV [68], we use 1024 as the hidden size of dense layers and the deterministic model state dimension.

**Reward predictors, actor, and critic.** Both of the dual reward predictors, as well as actor and critic, are all implemented as 4-layer MLPs with the hidden size 400 and ELU activation function, following DreamerV2 [23] and APV [68].

## C.4 Hyperparameters

For the newly introduced hyperparameter, we use $\beta_r = 1.0$ for all tasks. We set $\beta_z = 1.0$ for pre-training, and $\beta_z = 0.0, \beta_s = 1.0$ for fine-tuning, following APV [68]. Despite we adopt a deeper architecture for both image encoders and decoders, we find that hyperparameters are relatively robust to architectures, and the new deeper architecture under the same hyperparameters can match or even slightly improve the performance originally reported by APV. Thus, we use the same hyperparameters with APV. For completeness, we report important hyperparameters in Table 2.

## C.5 Qualitative Analysis Details

We explain the visualization scheme for Figure 8. For video prediction in Figure 8a, we sample a video clip of length 25 and let the model observe the first 15 frames and then predict the future 10 frames open-loop by the latent dynamics model. The context frame is selected as the last of the observed 15 frames. To visualize context attention, we compute the cross-attention weight across the $16 \times 16$ feature maps without random masking and plot a heatmap of the weights averaged over attention heads and over attention targets of a small region (for example, $3 \times 3$) on the decoder feature map. For video representations in Figure 8b, we sample video clips of length 25 and compute the averaged model states $\mathrm{avg}(z_{1:T})$ of dimension 2048 for each video clip, which are then visualized by t-SNE [74].

## C.6 Computational Resources

We implement all the methods based on PyTorch [54] and train with automatic mixed precision. In terms of parameter counts, ContextWM consists of 26M and 47M parameters for video pre-training and MBRL, respectively, while a plain WM consists of 24M and 44M parameters. In terms of training time, it takes ~37 hours for pre-training of ContextWM over 600K iterations, and fine-tuning of ContextWM with MBRL requires ~24 hours for each run of Meta-world experiments over 250K environment steps, ~23 hours for each run of DMC Remastered experiments over 1M environment steps and ~16 hours for each run of CARLA experiments over 150K environment steps, respectively. Although ContextWM introduces new components of the model, we find it does not significantly impact the training time, since the context encoder only needs to operate on one frame of observations, and the cross-attention between features also accounts for a relatively small amount of computation. In terms of memory usage, Meta-world experiments require ~18GB GPU memory, and DMC Remastered and CARLA experiments require ~7GB GPU memory, thus the experiments can be done using typical 24GB and 12GB GPUs, respectively.

Table 2: Hyperparameters in our experiments. We use the same hyperparameters as APV [68].

|  | Hyperparameter | Value |
|---|---|---|
| Pre-training from Videos | Image size | $64 \times 64 \times 3$ |
|  | Image preprocess | Linearly rescale from $[0, 255]$ to $[-0.5, 0.5]$ |
|  | Video segment length $T$ | 25 |
|  | KL weight $\beta_z$ | 1.0 |
|  | Optimizer | Adam [37] |
|  | Learning rate | $3 \times 10^{-4}$ |
|  | Batch size | 16 |
|  | Training iterations | $6 \times 10^5$ for SSv2 / Human / YouTubeDriving |
|  |  | $1.2 \times 10^6$ for Assembled dataset |
| Fine-tuning with MBRL | Observation size | $64 \times 64 \times 3$ |
|  | Observation preprocess | Linearly rescale from $[0, 255]$ to $[-0.5, 0.5]$ |
|  | Trajectory segment length $T$ | 50 |
|  | Random exploration | 5000 environment steps for Meta-world |
|  |  | 1000 environment steps for DMCR and CARLA |
|  | Replay buffer capacity | $10^6$ |
|  | Training frequency | Every 5 environment steps |
|  | Action-free KL weight $\beta_z$ | 0.0 |
|  | Action-conditional KL weight $\beta_s$ | 1.0 |
|  | Representative reward predictor weight $\beta_r$ | 1.0 |
|  | Intrinsic reward weight $\lambda$ | 1.0 for Meta-world |
|  |  | 0.1 for DMCR and CARLA |
|  | Imagination horizon $H$ | 15 |
|  | Discount $\gamma$ | 0.99 |
|  | $\lambda$-target discount | 0.95 |
|  | Entropy regularization $\eta$ | $1 \times 10^{-4}$ |
|  | Batch size | 50 for Meta-world |
|  |  | 16 for DMCR and CARLA |
|  | World model optimizer | Adam |
|  | World model learning rate | $3 \times 10^{-4}$ |
|  | Actor optimizer | Adam |
|  | Actor learning rate | $8 \times 10^{-5}$ |
|  | Critic optimizer | Adam |
|  | Critic learning rate | $8 \times 10^{-5}$ |
|  | Evaluation episodes | 10 for Meta-world and DMCR |
|  |  | 5 for CARLA |

# D   Additional Experimental Results

## D.1   Comparison with Additional Baselines

We compare our approach with a state-of-the-art model-free RL method DrQ-v2 [79] and a model-based RL method Iso-Dream [52]. We adapt their official implementations[78] to all the tasks in our experiments and present the results in Figure 9. As shown, our method consistently outperforms state-of-the-art baselines, which demonstrates the merit of pre-training from in-the-wild videos. Model-free methods such as DrQ-v2 have been shown to perform worse than model-based methods [68, 79], and our results are consistent with this. We also remark that our method differs from Iso-Dream [52] since we separate modeling of context and dynamics at the semantic level while Iso-Dream isolates controllable and noncontrollable parts at the pixel level. Thus, contextual information such as body color has to be modeled by the dynamics branch in Iso-dream, which probably results in worse decoupling of context and dynamics as well as inferior performance for challenging tasks. Moreover, compared to Iso-Dream, our model makes a simpler adjustment to Dreamer architecture.

---

[7]`https://github.com/facebookresearch/drqv2`
[8]`https://github.com/panmt/Iso-Dream`

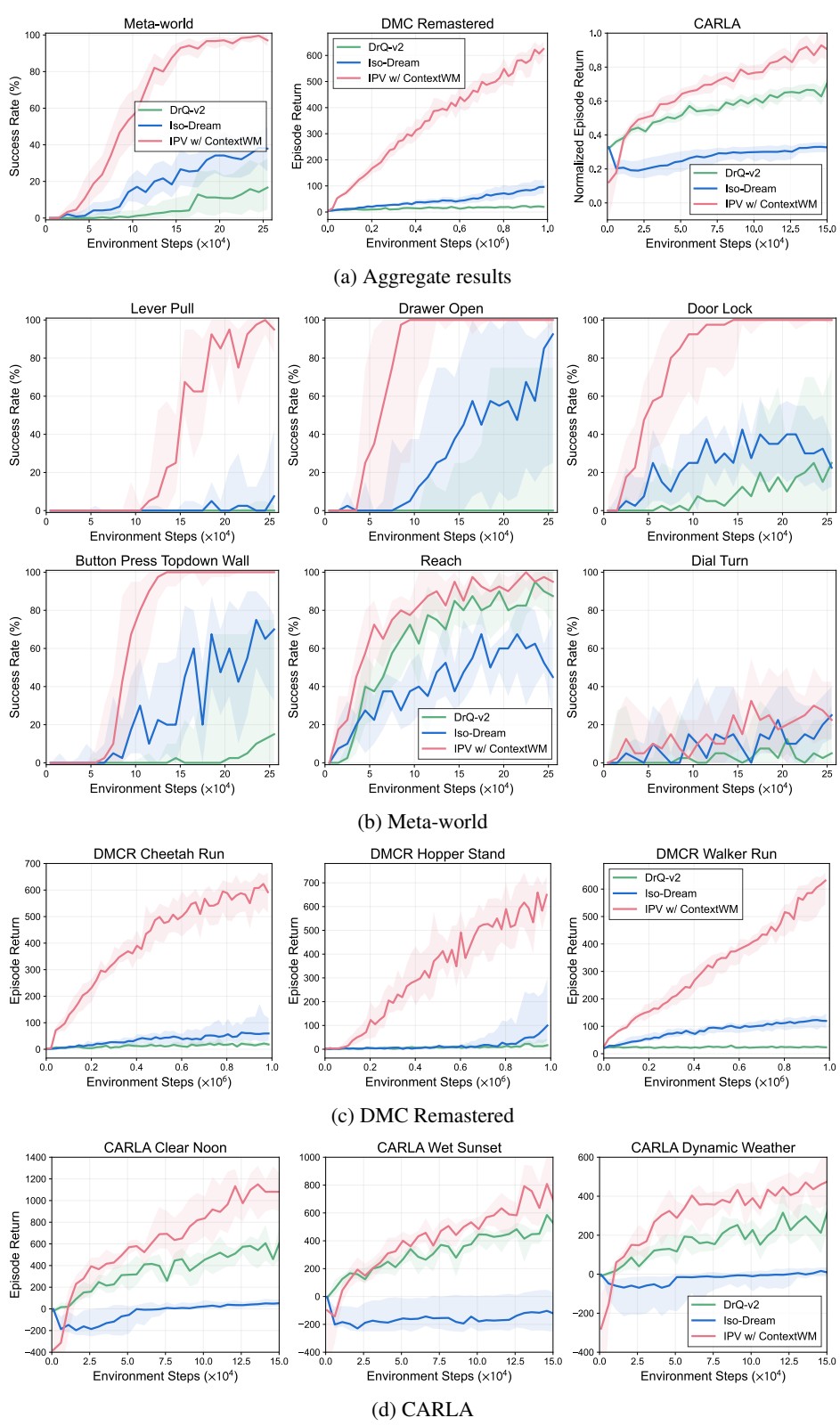

Figure 9: Comparison with additional baselines, DrQ-v2 [79] and Iso-Dream [52]. (a) Aggregate performance of different methods across all tasks on each domain. (b)(c)(d) Learning curves of different methods on each task from Meta-world, DMC Remastered, and CARLA, respectively. We report aggregate results across eight runs for each task.

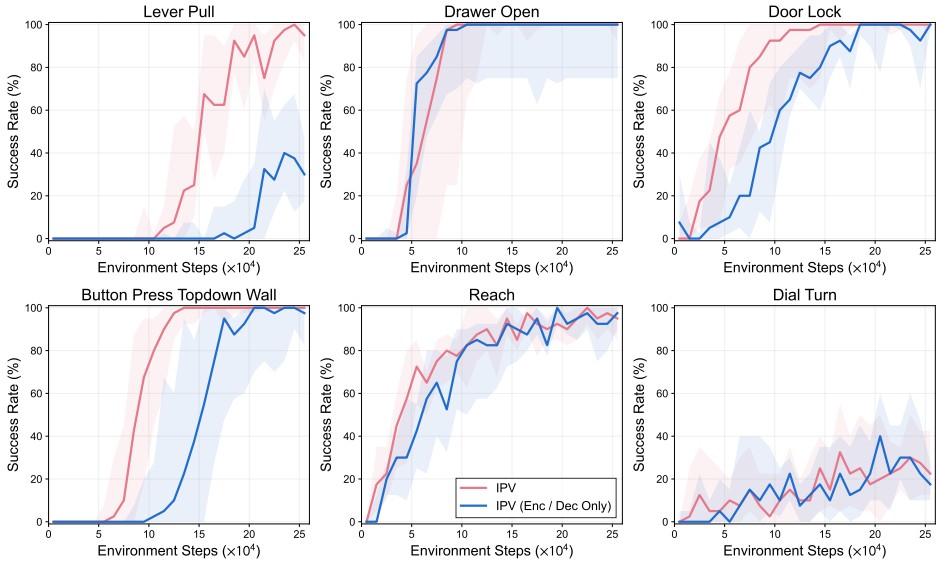

Figure 10: Learning curves of IPV with ContextWM on Meta-world manipulation tasks when only the pre-trained parameters of the visual encoders and decoder are transferred, i.e., without transferring dynamics knowledge obtained by pre-training (denoted by *Enc / Dec Only*). We report success rates, aggregated across eight runs for each task.

## D.2   Effects of Dynamics Knowledge

To investigate the importance of dynamics knowledge captured by pre-trained ContextWM, we report the performance of IPV with ContextWM when only the pre-trained parameters of the visual encoders and decoder are transferred. Figure 10 indicates that only transferring visual encoders and decoder performs worse on four of six tasks. These results align with the observation of APV [68] and highlight that pre-trained world models are capable to capture essential dynamics knowledge beyond learning visual representations solely [49, 53], further boosting visual control learning.

# E   Additional Discussions

**Particularly strong performance gains on DMCR tasks.**   Our method obtains considerable performance gains on DMCR tasks. The main reason is that DMCR is a purposefully designed benchmark, which measures visual generalization and requires the agent to extract task-relevant information as well as ignore visual distractors [19]. As demonstrated in Figure 8c, our ContextWM has the advantage of separately modeling contexts (task-irrelevant in DMCR) and dynamics (task-relevant in DMCR), which avoids wasting the capacity of dynamics models in modeling low-level visual details. Furthermore, pre-training with in-the-wild videos enables our models to eliminate diverse distractors and capture shared motions, which is essential for visual generalization in RL. In contrast, a plain WM needs to model complicated contexts and dynamics in an entangled manner, which adds difficulty to dynamics learning and behavior learning on these features. We also emphasize that, motivated by separating contexts and enhancing temporal dynamics modeling, our proposed IPV w/ ContextWM is a general-purpose framework and, as shown in our experiments, can obtain adequate performance gains on various benchmarks that have more complicated entangling of contexts and dynamics beyond DMCR.

**Discussion with Transformer-based world models.**   While a cross-attention mechanism is leveraged for context information conditioning, we still use an RNN-based latent dynamics model from Dreamer [21] to model history observations. More powerful sequential backbones such as Transformers are orthogonal to our primary technical contribution of explicit context modeling and conditioning. Nevertheless, exploring Transformer-based world models [10, 47, 58] is valuable, not only because of its favorable scalability but also of its flexibility to incorporate pre-training on action-free videos by simply changing conditioning variables and to condition the contextualized decoder in a more native way. Overall, the combination of our pre-training framework with a Transformer architecture holds great potential, and we will delve deeper into this aspect in future work.

# F  Additional Visualizations

**Video prediction.**  We present additional showcases of video prediction by plain WM and ContextWM, respectively, on three video datasets, in Figure 11. On the challenging Something-Something-v2 and YouTube Driving datasets, our model provides better prediction quality in comparison to a plain WM and well captures dynamics information (e.g., how objects are moving) with concentrated cross-attention. However, on the Human3.6M dataset, although our model better predicts human motions, we find its context cross-attention is relatively divergent. This result supports our conjecture that our model pre-trained on the Human3.6M dataset lacking in diversity may suffer from overfitting, which results in insufficient modeling of context and dynamics as well as inferior performance when fine-tuned with MBRL.

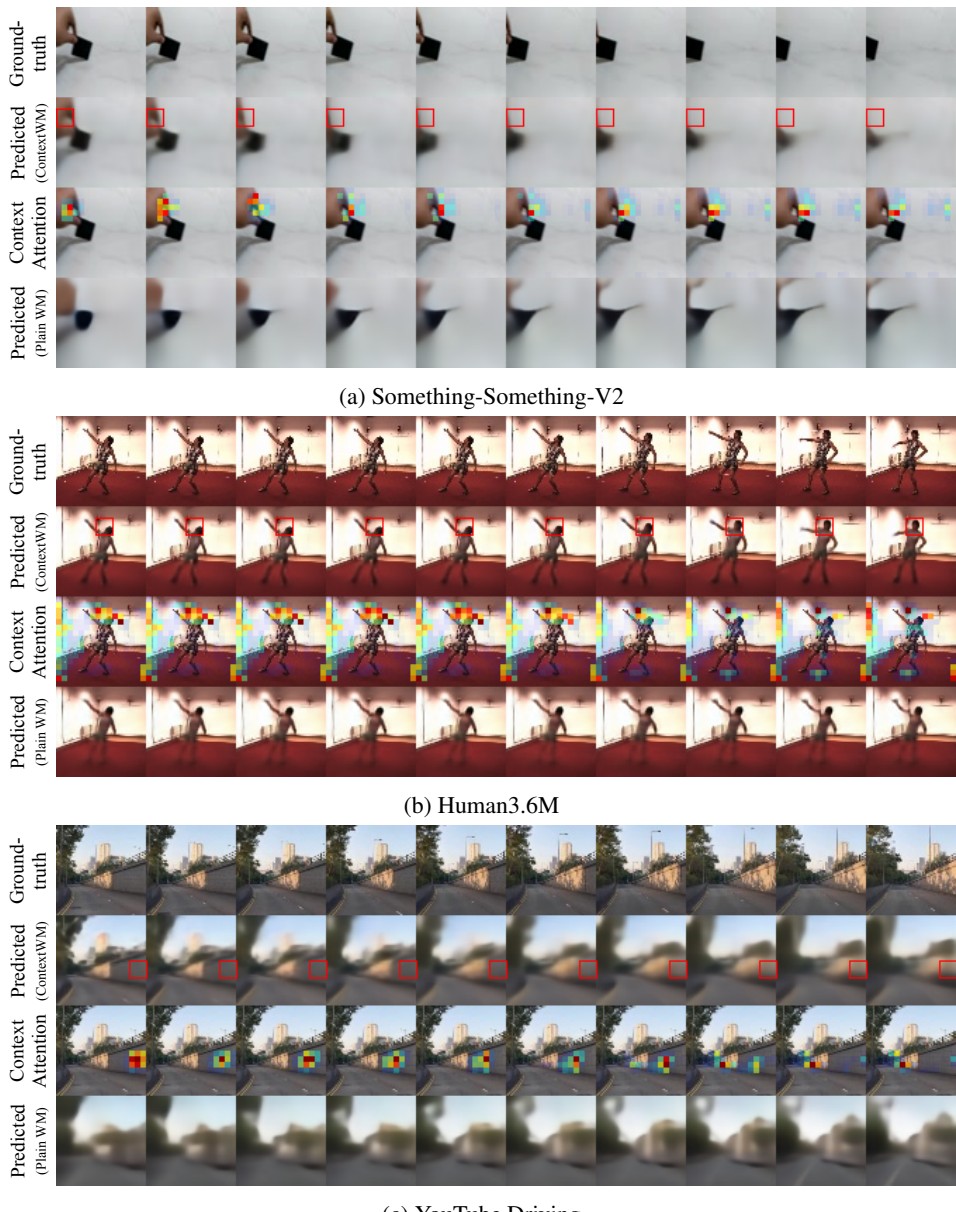

(a) Something-Something-V2

(b) Human3.6M

(c) YouTube Driving

Figure 11: Future frames predicted by plain WM and ContextWM, respectively, on three in-the-wild video datasets. We also visualize context attention weight from a target region of prediction (red box) to the context frame. We refer to Appendix C.5 for details.

**Video representation.** We present in Figure 12 additional t-SNE visualization similar to Figure 8. Our ContextWM learns well-distributed representations that clearly separate motion directions while plain WM fails.

**Compositional decoding.** Figure 13 includes additional compositional decoding analysis on the DMCR Walker Run task.

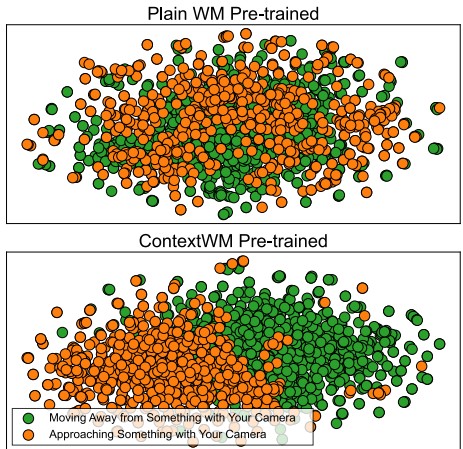

Figure 12: Additional t-SNE visualization of average pooled representations of SSv2 videos with two distinct labels, from a pre-trained plain WM and ContextWM, respectively.

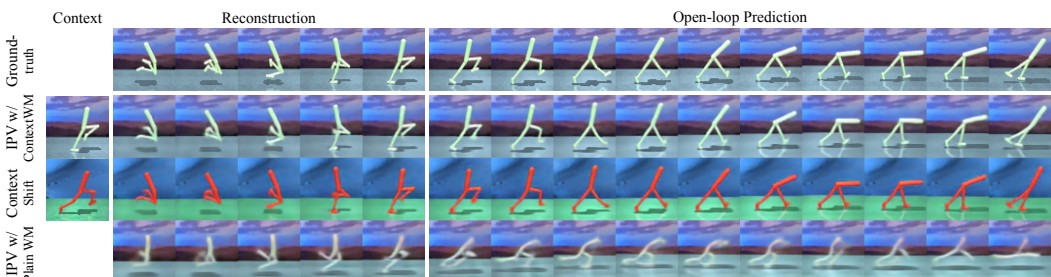

Figure 13: Additional compositional decoding analysis on the DMCR Walker Run task.

## G   Broader Impact

Advancements in world models and training them with in-the-wild videos can raise concerns about the generation of deepfakes, realistic synthetic videos that mimic real events or individuals. Additionally, advancements in MBRL can also increase automation in domains like robotic manipulation and autonomous driving, bringing efficiency and safety benefits but also leading to job displacement and socioeconomic consequences. However, it is important to note that as the field is still in its early stages of development and our model is only a research prototype, the aforementioned negative social impacts are not expected to manifest in the short term.

