# OpenReview forum: "Pre-training Contextualized World Models with In-the-wild Videos for Reinforcement Learning"
_NeurIPS.cc/2023/Conference — NeurIPS 2023 poster_

### Official Review · Reviewer_MYvK · 2023-06-25

**Soundness:** 2 fair
**Presentation:** 3 good
**Contribution:** 1 poor
**Rating:** 3
**Confidence:** 3

**Summary:**

This paper presents a method for learning world models from in-the-wild videos. By utilizing a context encoder to capture contextual information, the proposed method explicitly models both the context and dynamics to facilitate knowledge transfer across scenes. Experiments are performed on various simulation benchmarks such as Meta-world, DMC Remastered, DeepMind Control Suite, and CARLA.

**Strengths:**

The motivation for learning better world models by disentangling context information and dynamics is clear and seems reasonable to me;

The proposed method for learning context information is simple and straightforward;

The paper is well-organized and easily read.

**Weaknesses:**

The proposed framework is based on the assumption that the context information lies equally in each frame, however, it is very likely that some moving objects might be occluded at some time. There are no specific designs for handling these situations;

The experiments mainly validate the sample efficiency of the RL training process, there is no sufficient ablation study on the learned context information;

Compared to the vanilla WM baseline, the performance of RL training on the Meta-world benchmark seems not very impressive;

The predicted video frames seem not very promising;

There is no discussion of limitations and failure cases.

**Questions:**

The authors propose to randomly select a frame for predicting the context feature, have the authors tried to use a fixed frame (e.g., the first frame or the last frame of the video clip)? Would the context feature extracted from the different frames be consistent with each other? And how would choosing different frames affect the final performance of the RL training?

It is assumed that the learned context feature captures the static properties of objects (e.g., colors, shapes), would modifying the context feature allow us to generate diverse video frames (e.g., changing the color of the object)? It would be interesting to show more temporal consistent video frames by editing the context feature.

**Limitations:**

There is no discussion of limitations and failure cases.

---

> ### Author Rebuttal · Authors · 2023-08-10
>
> Many thanks to Reviewer MYvK for providing insightful comments and questions.
>
> **Q1**: Discussion and ablation study on context frame selection
>
> **Single vs. multiple**: We agree with the reviewer that, in general, a single context frame cannot provide perfect contextual information, and it is challenging to learn fully disentangled contextual and dynamics representations. But our intuition is that **propagating most of the contextual information through a separate encoder can facilitate shared dynamics knowledge transfer between visually distinct pre-training and downstream domains**. Important dynamics information, such as occluded moving objects, can also be captured by the dynamics branch, as RSSMs can handle partial observability. As a first step towards broadly applicable world model pre-training, our experiments in various domains support that a single context frame can overcome complicated context distribution to unlock positive transfer. We have discussed the limitations and future work to utilize multiple frames in more complicated tasks (see $\underline{\text{Q1 in the global response}}$).
>
> **Feature consistency**: Our random selection strategy can encourage consistency between context features extracted from different frames. We note that they are not required to be strictly consistent, as our cross-attention mechanism allows contextual information to propagate adaptively between different frames (see $\underline{\text{Fig.8 of main paper}}$ for example).
>
> **Experiment results**: To support the discussion, we have experimented using a fixed frame (the first and last one) and multiple frames (randomly sampled three frames concatenated as inputs of the contextual encoder). Results are presented in **$\underline{\text{Fig. 3 of the global response attachment}}$**. We conclude that **different context frame selection schemes do not significantly affect performance**. Utilizing multiple frames does not provide benefits, probably because the experimental environment is simple or the way incorporating multiple frames is crude.
>
> **Q2**: Predicted video frames seem not very promising
>
> We note that SSv2 and YoutubeDriving datasets are large-scale in-the-wild datasets. Generating high-fidelity videos on these domains is especially difficult, and little literature has successfully achieved this. Developing generative models of videos is still a rapidly evolving and immature field.
>
> However, as pointed out at the beginning of the paper, through large-scale video pre-training of a world model, **our work aims to boost sample efficiency of downstream model-based RL, rather than fidelity of video prediction**. In order to learn beneficial representation for MBRL, we believe world models should focus on essential dynamics information, such as object positions and motions, instead of low-level visual details. This naturally motivates our decision-oriented design facilitating separate contexts and dynamics modeling. We present predicted video frames in $\underline{\text{Fig. 8 of main paper}}$ in order to illustrate that ContextWM can learn important dynamics features from large-scale pre-training, which benefits downstream MBRL. On simpler datasets, e.g., Human3.6M, our model can make much better predictions. However, as shown in $\underline{\text{Fig. 7b of main paper}}$, pre-training on this dataset lacking in diversity can even hurt our ultimate goal: sample efficiency of visual control learning. Advances in generative models can help develop stronger backbones for world models but are not the focus of this work and are orthogonal to our contributions.
>
> **Q3**: Generating diverse video frames
>
> Thanks for the insightful question. We have experimented with modifying the context feature to generate novel videos. We use the DMCR domain as our testbed as it supports modifying visual factors. While it is difficult to manipulate context features directly, we have done a workaround by sampling a frame from another trajectory to extract contextual information. In this way, we can modify the agent's color and the background's texture. As shown in $\underline{\text{Fig. 1 of the global response attachment}}$, ContextWM can make temporal consistent video predictions by correctly combining the new contextual information with the original dynamics information. For further details, see the qualitative analysis part of $\underline{\text{Q2 in the global response}}$.
>
> **Q4**: Performance gain on Meta-world
>
> Following prior work, Meta-world performance is measured by the success rate of 10 episodes, which naturally has a high variance. We have made a great effort to solidify our results, including massive repeated experiment runs. In $\underline{\text{Fig. 5b of main paper}}$, we have demonstrated the **statistical significance of our improvement over vanilla WM with a clear margin, aggregated over 48 runs on six Meta-world tasks**, following the protocols of APV and Agarwal et al. [1]. For particular tasks, e.g., Drawer Open, our method can learn with only half of the environment interactions, compared to the baseline.
>
> [1] Agarwal et al. Deep reinforcement learning at the edge of the statistical precipice. NeurIPS, 2021.
>
> **Q5**: Limitations and failure cases
>
> We apologize for the limited discussion of limitations. We will add a detailed discussion in a future revision. Please see $\underline{\text{Q1 in the global response}}$ for the revised discussion on limitations.
>
> For failure cases, we have observed that it does not always provide significant gains when pre-training our model on video datasets from different domains or of different amounts (e.g., Human3.6M). We have discussed possible reasons in corresponding paragraphs in the experimental section.

---

> > ### Comment · Reviewer_MYvK · 2023-08-18
> > **POST-REBUTTAL**
> >
> > I acknowledge the authors' dedication to incorporating the updates. Following a meticulous examination of the rebuttal materials, my reservations regarding the modest advancement in performance and the absence of profound technical contributions persist. Given these considerations, I am leaning toward upholding my initial evaluation.

---

> > > ### Author Response · Authors · 2023-08-19
> > > **Response to Post-rebuttal Feedback by Reviewer MYvK**
> > >
> > > Dear Reviewer MYvK,
> > >
> > > Thank you again for your time and effort in reviewing our paper. We appreciate your careful review of our rebuttal materials and your recognition of our efforts in incorporating updates.
> > >
> > > We recognize and respect the diverse perspectives regarding the significance of a paper. However, due to the dramatic inconsistency between our opinions, we **kindly request your reconsideration of your reservations on the advancement in performance and technical contributions**. We want to highlight that in accordance with the [NeurIPS 2023 Reviewer Guidelines](https://neurips.cc/Conferences/2023/ReviewerGuidelines), we need specificity, flexibility, and timeliness in your reviews in order for us to better address your concerns.
> > >
> > > While we've endeavored with full-time efforts to address your concerns, unfortunately, we've observed that your first feedback logged on 10 hours ago is somewhat vague and limited in specificity and evidence. Here, we will provide further responses to address your concerns, hopefully to your satisfaction, and to help the other reviewers understand the opinions from both sides.
> > >
> > > **(1)** Performance advancement
> > >
> > > Please kindly refer to our $\underline{\text{Q4 in our rebuttal}}$ above and our further clarification below.
> > >
> > > We have made a great effort to support the statistical significance of our improvement and compared our method with typical baselines from previous RL literature, including **DreamerV2 in our main paper, DrQ-v2/Iso-Dream in our supplementary material, and DreamerV3/TransDreamer in our rebuttal**. Our results consistently demonstrate the superior efficacy against these typical RL baselines across various domains and tasks, showing the benefits of our in-the-wild pre-training (IPV) framework and the contribution to the RL community.
> > >
> > > Regarding improvement upon our most relevant baseline APV [1] (named as 'IPV w/ vanilla WM' in our paper), it is still statistically significant in Fig. 5b of our paper, aggregated across 48 runs over six tasks of Meta-world. Note that the improvements of  'ContextWM (Pre: O)' against  'ContextWM (Pre: X)' and 'vanilla WM (Pre: O)' are **of a comparable magnitude with improvements made by previous publications** (for example, 'APV (Pre: O / Int: O)' against 'APV (Pre: X / Int: O)' in Fig. 6b of APV paper [1]). While APV makes this improvement with a domain-specific pre-training, we utilize more broadly applicable in-the-wild pre-training.
> > >
> > > **(2)** Technical contributions
> > >
> > > Please kindly refer to our $\underline{\text{Q2 in our rebuttal}}$ above and we apologize that we did not sufficiently state our contribution in the rebuttal for you.
> > >
> > > As stated, **our major technical contribution is to unleash the power of in-the-wild pre-training from videos to boost the sample efficiency of downstream MBRL**. Making world models benefit from in-the-wild pre-training is a critical precondition to scale up to big data and large models since it provides world knowledge widely generalizable and applicable to various downstream tasks. As Reviewer *wPqn* pointed out, 'learning world models on in-the-wild videos is hard', and we highlight that **no previous work has demonstrated positive transfer of a world model from in-the-wild videos** (see Fig. 8c of APV paper [1]). Motivated by the intricate property of in-the-wild contexts, we propose Contextualized World Models, a framework to explicitly separate contextual information and encourage shared dynamics modeling. Our experiments support that **our model successfully breaks the transfer barrier**.
> > >
> > > Overall, we have systematically studied **a new problem** (IPV, in-the-wild pre-training from videos), proposed **a new method** tailored for this problem (ContextWM), and demonstrated **significant performance gain** across various domains, which we believe all contribute to the community and help pave the path ahead toward general world models.
> > >
> > > [1] Seo, Y., et. al. Reinforcement learning with action-free pre-training from videos. ICML 2022.
> > >
> > > We hope that these responses can address your issues and shed light on the significance and solidity of our work. Could you please consider re-evaluating our work based on the updated information? We remain eager to address any lingering concerns and value an open and interactive discussion. Looking forward to your reply.
> > >
> > > Best regards,
> > >
> > > Authors.

---

> ### Comment · Area_Chair_NT8x · 2023-08-17
> **Your (reviewer) response to the author rebuttal is missing. Please do it ASAP.**
>
> Dear Reviewer,
>
> The author has posted their rebuttal, but you have not yet posted your response. Please post your thoughts after reading the rebuttal and other reviews as soon as possible. All reviewers are requested to post this after-rebuttal-response.

---

> ### Author Response · Authors · 2023-08-21
> **Discussion period ends soon**
>
> Dear Reviewer MYvK,
>
> As the Reviewer-Author discussion period concludes soon, we kindly request your feedback on our rebuttal and post-rebuttal response. **We've earnestly addressed your concerns about performance advancement and technical contributions**.
>
> We appreciate your feedback on whether our responses meet your expectations. If any concerns remain, we're eager for further discussion. If you find our responses satisfactory, we hope for your reconsideration in assessing our paper.
>
> Thank you for your valuable time and consideration. We anticipate your response.
>
> Best regards,
>
> Authors

---

### Official Review · Reviewer_wPqn · 2023-07-05

**Soundness:** 4 excellent
**Presentation:** 4 excellent
**Contribution:** 3 good
**Rating:** 7
**Confidence:** 4

**Summary:**

Learning a world-model that can generalize to different domains and tasks is difficult. The authors enhanced an existing framework for pre-training world models using in-the-wild videos, which can be fine-tuned on downstream tasks. In particular, the authors introduce a contextual encoder which helps in disentangling temporal dynamics from static contexts. Additionally, they include a cross-attention mechanism and dual reward predictors to improve the learning of task-relevant representations.

**Strengths:**

The manuscript is very well written and structured.

●	The idea of using a context to encode static information is novel, well-motivated and it shows good results in the DMC remastered task

●	Learning world models on in the wild videos is hard (and so far does not help downstream tasks, as shown by Seo et al. [49]) - The extension of the Action-Free Pre-training from Videos approach to in-the-wild videos is convincing and the context modulation via a U-Net for reconstruction is innovative.

●	The authors perform several relevant ablations to illustrate the role of the different proposed components.


**Weaknesses:**

Overall, the context is only used together with the latent dynamic to decode the image. Is there a clear reason why the context is not taken into account for predicting the reward (in Fig. 3b the arrow of the context that goes to the reward seems to be misleading)? As the authors stated, the context might implicitly encode some important information about the task, e.g., the static position of an object, which can be helpful during the fine-tuning phase.

**Questions:**

●	What does the context encode? The motivation and the qualitative analysis are convincing, but it would also be useful to test this further. This could be done via a decoding analysis. In particular, it would be very interesting to see the difference between the context and the dynamic in the case of the DMC remastered task, where the contextWM provides a significant advantage.

●	Why do you think is the gap in performance to prior methods particularly strong for DMCR?

●	Looking at figure 6b (bottom), the performance of the pre-trained ContextWM on the SSv2 dataset for the CARLA driving task seems to outperform the one that is not-pretrained. However, the effects of the dataset domain of figure 7b (right) shows that there is almost no difference between the one pre-trained on SSv2 and the one without pre-training. What am I missing?

●	Does the contextWM help to achieve better generalization? Could one, e.g., purposefully change the color, size or shape of the object for the meta-world and achieve better performance than the WM?


**Limitations:**

Fine

---

> ### Author Rebuttal · Authors · 2023-08-10
>
> We sincerely thank Reviewer wPqn for providing a detailed review and insightful questions.
>
> **Q1**: Utilizing contexts for predicting the reward
>
> We agree that, in general, contextual and dynamics information are both important for task-relevant predictors (reward predictor, actor, and critic). We can also design cross-attention mechanisms for these MLP predictors. However, since contextual information in our architecture has a complicated structure (multi-scale CxHxW feature maps), this may bring extra design and implementation efforts, additional hyperparameters, and computation costs. To maintain a simple architecture and fair comparisons, we opt to predict rewards with only dynamics features and utilize our proposed dual reward predictor structure to encourage completely task-relevant feature encoding, which has been shown to work well in experimental benchmarks. Incorporating contexts and dynamics information for reward prediction and behavior learning in more complicated tasks is a promising future direction.
>
> For the misleading Fig. 3b, we will revise it to make it clearer in a future revision. Thanks for your valuable suggestion.
>
> **Q2**: Visualization in the case of the DMCR tasks
>
> Thanks for the valuable suggestion of a decoding analysis on the DMCR domain. To demonstrate the difference between context and dynamics, we conduct a **compositional decoding analysis** by sampling a random frame from another trajectory to replace the original context and leave dynamics unchanged. Our ContextWM shows excellent compositionality as it can correctly combine the new contextual information with the original dynamics information. We conclude that in this domain, the context encodes static visual factors such as the agent's body color, the background's texture, etc. For further details, please refer to the qualitative analysis part of $\underline{\text{Q2 in the global response}}$.
>
>
> **Q3**: Particularly strong performance on DMCR tasks
>
> Thanks for your insightful question. Please refer to $\underline{\text{Q2 in the global response}}$ for the detailed response.
>
> **Q4**: Effects of pre-training with SSv2 on CARLA
>
> We apologize for a mistake of plotting: **the 'w/o Pre-train' curve in Fig. 7 (CARLA) should be the same as the 'ContextWM (Pre: X)' curve in Fig. 6 (bottom)**, but was incorrectly plotted as the 'vanilla WM (Pre: O)' curve in Fig. 6 by mistake. We have carefully checked the figures to ensure there are no other mistakes and will correct this in a future revision—many thanks for pointing it out.
>
> **Q5**: ContextWM promotes generalization
>
> We believe that our ContextWM can promote generalization due to better design of context and dynamics modeling. While changing the color, size, or shapes in Meta-world is difficult, we have conducted similar experiments in another benchmark, DMC Remastered, which randomly resets all visual factors on the initialization of each training and evaluation episode. As shown in $\underline{\text{Fig. 6 of main paper}}$, ContextWM outperforms vanilla WM significantly on DMCR, both with or without pre-training. These results demonstrate that **ContextWM can achieve better generalization and performance in unseen visual environments**. Measuring out-of-distribution generalization ability (e.g., training on standard DMC/Meta-world and testing on visually modified ones) of ContextWM and vanilla WM is an interesting future direction.

---

> > ### Comment · Reviewer_wPqn · 2023-08-14
> >
> > Thank you for the detailed rebuttal to the comments of the other reviewers as well as mine. I think the paper was already good and also improved during the rebuttal period. Thus, I maintain my accept score (7). I hope the other reviewers, who had lower scores, can check out the response!

---

> > > ### Author Response · Authors · 2023-08-14
> > > **Appreciation for Your Support**
> > >
> > > Dear Reviewer wPqn,
> > >
> > > Your support and maintained acceptance score are sincerely appreciated. Thank you for recognizing our improvements and suggesting that other reviewers check our responses.
> > >
> > > Best regards,
> > >
> > > Authors

---

### Official Review · Reviewer_ajER · 2023-07-07

**Soundness:** 2 fair
**Presentation:** 4 excellent
**Contribution:** 2 fair
**Rating:** 7
**Confidence:** 4

**Summary:**

This paper studies whether large-scale in-the-wild datasets can be used to pre-train world models for efficient downstream reinforcement learning. Specifically, they introduce Contextualized World Models (ContextWM), an architecture specifically designed to learn to separate context and dynamics modeling. Their experimentation reveals that the proposed methodology outperforms DreamerV2 on a variety of downstream tasks.

**Strengths:**

- Well-written. The paper is well-written, and the figures aid in the understanding of the methodology.
- Variety of experiments. The authors conduct a variety of experiments, comparing not just to DreamerV2, but also analyzing the effects of pre-training, architecture choices, dataset domain, etc. Furthermore, both quantitative and qualitative comparisons are included.
- Strong experimental. Relative to DreamerV2, the proposed methodology achieves a strong performance -- the gap seems to be particularly large for the DMC Remastered tasks.

**Weaknesses:**

- Missing comparison to prior work. APV [49] is the most similar prior work which this work builds off of (and seems to be the SOTA in this space), and yet the proposed method is not compared to APV. Many of the tasks used, hyper-parameters, and evaluation protocols adopted are from APV, allowing for a comparison, yet somehow, this comparison is omitted. Looking at the results plots in the APV paper, visually, the performance of ContextWM seems similar to that of APV (and in some cases clearly worse, e.g. in the dial turn task). While one may claim that a comparison to APV may be unfair because both the data and the model would be different, this would still reveal whether the ability to use a larger amount of in-the-wild data as well as the changes to the model architecture are actually beneficial. Furthermore, training APV on this large-scale data and seeing whether ContextWM outperforms it would be an experiment which would reveal whether the architectural changes proposed are significant. Finally, APV primarily compares to DreamerV2 as this was the SOTA when APV was published -- now, DreamerV3 seems to be the SOTA, so a comparison to DreamerV3 should be conducted.
- Incomprehensive ablations. It is not clear which one environment the ablation study is conducted on. It would be a lot more convincing if the ablation study was done across tasks, and if the trends held true across tasks as this would alleviate concerns of cherry-picking a task.

**Questions:**

- Is it possible to qualitatively demonstrate that ContextWM is able to separate context and dynamics modeling?
- For some tasks, pre-training and choice of data to pre-train with makes a huge difference, e.g. DMC Re-mastered. For others, not so much, e.g. CARLA. Is there a sense of why this is the case?

**Limitations:**

There is no discussion on the limitations of the work.

---

> ### Author Rebuttal · Authors · 2023-08-10
>
> Many thanks to Reviewer ajER for providing a thorough review and valuable questions.
>
> **Q1**: **Comparison with APV**
>
> We **respectfully disagree with the comments that we do not compare with APV**. We apologize for not clarifying that **our 'IPV w/ vanilla WM' in $\underline{\text{Fig. 5 and 6 of main paper}}$ is the APV baseline trained on the same data**, equipped with the stacked latent model and intrinsic bonus proposed by APV. We brand it with the new name to emphasize that it is pre-trained on **I**n-the-wild data rather than only **A**ction-free data. Furthermore, since neither the one-layer RSSM in Dreamer nor the stacked RSSM in APV has a contextualized component as ContextWM, we named them vanilla WM at the beginning of our experimental section. We will change to more proper names in a future revision. In a word, **we have already trained APV on large-scale data and demonstrated that the architectural changes proposed are significant**.
>
> We also note that the performance gap with the original APV comes from different pre-training datasets. **Using the same curated RLBench dataset, our ContextWM (see $\underline{\text{Fig. 7b of main paper}}$) can outperform originally reported APV results.** (Interestingly, we also find that the dial turn task can only benefit from RLBench data, regardless of the architecture.) It is unsurprising that pre-training data from a similar domain can further benefit downstream tasks. Nevertheless, **our methodology contribution unleashes the power of diverse video datasets instead of curated domain-specific ones to enable general-capable world model pre-training**. While the SSv2 dataset does not help downstream tasks in the original APV, our ContextWM pre-trained with it has been shown to benefit various control tasks.
>
> **Q2**: Comparison with DreamerV3
>
> Since our ContextWM is built upon DreamerV2, it is natural to compare it with DreamerV2 to reveal the significance of in-the-wild pre-training and the proposed architecture. **Our technical contributions are orthogonal to specific model-based RL methods** and can also combine with DreamerV3 to further improve performance, which is left for future work. Nevertheless, we conduct preliminary comparisons to DreamerV3 without pre-training. Results are presented in $\underline{\text{Fig. 4 of the global response attachment}}$. We conclude that even with several improved training techniques, DreamerV3 is still inferior to our method, showing the significance of in-the-wild video pre-training and explicit context modeling.
>
> **Q3**: **Clarification on ablation study**
>
> Following the protocol of Agarwal et al. [1] and APV,  **we conducted the ablation study on all the tasks and reported aggregated results**. Explanations of our ablation study results can be found in  $\underline{\text{Sec. 5.1}}$ and the captions of $\underline{\text{Fig. 5, 6, 7}}$. We will clarify it further in a future revision.
>
> [1] Agarwal et al. Deep reinforcement learning at the edge of the statistical precipice. NeurIPS, 2021.
>
> **Q4**: **Qualitative evaluation**
>
> While it is challenging to learn fully separated representation, we have provided qualitative evaluation in $\underline{\text{Sec. 5.5 and Fig. 8 of main paper}}$ to demonstrate the ability of ContextWM to separate contexts and dynamics. We also provide additional demonstrations in **$\underline{\text{Fig. 1 of the global response attachment}}$** to show that our model finetuned on the DMCR domain successfully learned disentangled representations of contexts and dynamics. For further details, please refer to the qualitative analysis part of $\underline{\text{Q2 in the global response}}$.
>
> **Q5**: Particularly significant performance gains on DMCR tasks
>
> Thanks for your insightful question. Please refer to $\underline{\text{Q2 in the global response}}$ for the detailed response.
>
> **Q6**: Limitations
>
> We apologize for the limited discussion of limitations. We will add a detailed discussion in a future revision. Please see $\underline{\text{Q1 in the global response}}$ for the revised discussion on limitations.

---

> ### Comment · Area_Chair_NT8x · 2023-08-17
> **Your (reviewer) response to the author rebuttal is missing. Please do it ASAP.**
>
> Dear Reviewer,
>
> The author has posted their rebuttal, but you have not yet posted your response. Please post your thoughts after reading the rebuttal and other reviews as soon as possible. All reviewers are requested to post this after-rebuttal-response.

---

> ### Author Response · Authors · 2023-08-19
> **Request of Reviewer's attention and feedback**
>
> Dear Reviewer ajER,
>
> Thanks again for your dedication to reviewing our paper.
>
> We write to kindly remind you that this is the last few days of the Reviewer-author discussion period. We have made every effort to address the concerns you suggested and improve our paper:
>
> - We **clarify that the 'IPV w/ vanilla WM' in the main paper is the APV baseline** trained on the same data as ours. Thus we have already trained APV on large-scale data and demonstrated that our architectural changes are significant.
> - We **provide additional comparison with DreamerV3**, where our model still performs the best.
> - We **clarify that the ablation study is conducted on all the tasks** and results are reported in aggregated forms. Ablation across all tasks show a consistent improvement of our model against baseline methods.
> - We **provide additional qualitative evaluation results** in the global response to show that our model is able to separate context and dynamics modeling.
> - We **explain the reason behind particularly significant performance gains** on DMCR tasks.
>
> Please kindly let us know if you have any remaining questions. If our responses have addressed your concerns, would you please consider re-evaluating our work based on the updated information? Looking forward to your reply.
>
> Sincerely,
>
> Authors

---

> ### Author Response · Authors · 2023-08-21
> **Discussion period ends soon**
>
> Dear Reviewer ajER,
>
> On this last day of the Reviewer-Author discussion period, we respectfully extend a final request for your valuable feedback on our rebuttal. Your perspective would greatly contribute to the thorough evaluation of our work.
>
> **Taking your suggestions, particularly regarding the experiments**, we believe that we have made a great effort to provide all the experiments and clarifications that we can. If our rebuttal has addressed your concerns, we hope the reviewer will reconsider the evaluation of our paper. We remain open to any further discussions.
>
> We sincerely extend gratitude for your dedicated review efforts and anticipate your response.
>
> Best regards,
>
> Authors

---

> > ### Comment · Reviewer_ajER · 2023-08-21
> >
> > Thank you for clarifying that the 'IPV w/ vanilla WM' vanilla baseline in the main paper is actually APV. This addresses my main concern that the proposed methodology was not fairly compared to prior work. The reviewer also agrees with the comments about DreamerV3 being orthogonal to the contribution of this work.
> >
> > Given that the misunderstanding has been resolved, and as the rebuttal has adequately addressed all of my concerns, I am increasing my rating to 7: Accept.

---

> > > ### Author Response · Authors · 2023-08-21
> > > **Appreciation for Your Feedback and Support**
> > >
> > > Dear Reviewer ajER,
> > >
> > > We sincerely appreciate your thoughtful re-evaluation of our paper and the subsequent rating adjustment. Your recognition of our contributions and the resolution of misunderstandings greatly encourage us. Your valuable input has undoubtedly enhanced the quality of our work.
> > >
> > > Thank you for your dedicated engagement and support.
> > >
> > > Best regards,
> > >
> > > Authors

---

### Official Review · Reviewer_JHvL · 2023-07-10

**Soundness:** 3 good
**Presentation:** 3 good
**Contribution:** 3 good
**Rating:** 5
**Confidence:** 4

**Summary:**

This paper proposes a Contexturelized World Model with In-the-wild Video Pretraining, which extends recently proposed action-free pre-training from videos (APV) to the case of contextualized video-prediction models. Specifically, they propose to sample a randomly chosen frame and use it as "contextualized information" for the prediction model. The "contextualized" information is combined into the prediction model via multi-scale cross-attention mechanisms. Besides, during the model-based RL phase, they propose to predict both pure task reward and the sum of the task reward plus a weighted intrinsic reward. Empirical validations on several well-known benchmarks (Meta-world, DMC, and CARLA) show the superior performance of the proposed method.

**Strengths:**

1. The paper is generally well-written and easy to follow.
2. While the proposed method is simple, experimental results show a stable performance gain by the proposed methods.
3. Besides the main contribution, the paper also provides an interesting analysis of the choice of pre-training datasets.

**Weaknesses:**

1. Technical novelty of the proposed method is not high. The main proposal is to add contextualized information to facilitate a better prediction. Besides, the paper does not discuss how the proposed approach (random sample selection) incorporates contextualized information well.

Especially, I'm not sure we can call the random variable $c$ a contextualized vector since it is computed via only single frames rather than the full context of the trajectories. In this sense, I'm wondering what happens if we predict/reconstruct observations using the same multi-scale cross-attention architecture but with the image from the same time step for each frame. In the setup, the model does not use contextualized information but has similar architecture to the proposed method, and thus more appropriate for the baseline.

2. Lack of discussion with transformer-based world models. As shortly discussed, several studies incorporate transformer architecture in the prediction model. Since the transformer architecture predicts the future via autoregressive fashion, I think it is more natural to handle the contextualized information. However, the current manuscripts lack discussion on this point and comparison with transformer-based world models

**Questions:**

1. Did you try methods other than random sample selection to compute contextualized information?

2. Did you try a comparison with transformer-based architecture?

3. The logic behind several sentences is hard to capture.  Could you please explain more about these sentences?

- In line 182, "Nevertheless, ..... Therefore, we propose a dual reward predictor" => Why does the dual predictor resolve the issue? Why not just balance lambda depending on the task?
- In line 234, "indicating that the performance gain of IPV with vanilla WM is primarily due to the intrinsic exploration bonus. " => The logic is unclear.

[Minor]
- In eq. 6, How do you compute KL when t=1?
- In Fig. 5-b, Pre: O should be Pre: ✓ or with Pre or something like that.
- If I correctly understood, Fig. 3 is a bit misleading as a context variable $c$ is directly input into each frame, rather then recurrent prediction.

**Limitations:**

A limitation section should be added.

---

> ### Author Rebuttal · Authors · 2023-08-10
>
> Many thanks to Reviewer JHvL for providing an insightful review and valuable comments.
>
> **Q1**: **How we incorporate contextual information**
>
> **Clarification**: We apologize for using the ambiguous term 'context' without elaborate clarification. Videos and visual control trajectories are **spatiotemporal** data. They have contexts in the temporal dimension (namely the history) and contexts in the spatial dimension (namely visual details, e.g., colors, shapes, and layouts of objects). 'Contexts' in our paper stands only for **spatial contexts**. A context frame is selected to extract (static) contextual information in a multi-scale manner, which encourages the latent dynamics model to focus on temporal dynamics instead of wasting model capacity on capturing low-level visual details. It is prevalent in the literature to utilize a reference image to condition generative models of videos [1, 2]. For **temporal context** modeling, we use standard RSSMs from Dreamer to model history observations. More powerful sequential backbones such as Transformers can be explored, but it is orthogonal to our technical contribution of explicitly modeling (spatial) contexts.
>
> **Novelty**: Reviewer kJR6 and wPqn both recognize our methods as innovative and well-motivated. To the best of our knowledge, we are the first to separately model contextual information for world models to handle complicated spatiotemporal data.
>
> **Empirical evidence**: We have shown that our model successfully unleashes the power of in-the-wild video pre-training and obtains significant gains on downstream tasks. Qualitative evaluation in $\underline{\text{Fig. 8 of main paper}}$ and $\underline{\text{Fig. 1 of the global response attachment}}$ also supports that our model can separate contexts and dynamics.
>
> **Reviewer-proposed baseline**: If understood correctly, the proposal from the reviewer, which uses the image from the same step as the context for each frame, can hardly learn useful representations for MBRL. Note that we extract multi-scale shortcuts from the context frame in a U-Net manner. Learning to reconstruct each frame conditioned on one fixed context encourages the model to learn temporal variations. However, when conditioned on the context frame from the same step for each frame, the model can learn to trivially copy low-level feature shortcuts for reconstruction. Our experiments of this baseline support our justification, where **the norm of RSSM features rapidly shrinks to zero during training**.
>
> [1] Singer et al. Make-a-video: Text-to-video generation without text-video data.
>
> [2] Esser et al. Structure and content-guided video synthesis with diffusion models.
>
> **Q2**: Other context frame selection methods.
>
> As shown in $\underline{\text{Fig. 3 of the global response attachment}}$, we have experimented with the first or the last frame as the context but found no significant performance difference. We have also tried randomly selecting three frames as the context and still obtained no performance gain, which indicates that a single frame is adequate for context modeling in our experimental benchmarks.
>
> **Q3**: Discussion and comparison with transformer-based architecture
>
> As discussed in Q1, transformer-based architecture is immature and **orthogonal** to our technical contribution. A combination of powerful transformer architecture and our framework has the potential to improve performance further, which is left for future work.
>
> Nevertheless, we have conducted preliminary comparisons to a transformer-based method, TransDreamer [3], without pre-training. Results are presented in $\underline{\text{Fig. 4 of the global response attachment}}$. We observe that even equipped with transformers, TransDreamer performs similarly to Dreamer and is inferior to our method. We do not compare with recently published work (IRIS, NeurIPS 2022 and TWM, ICLR 2023) since they only support discrete control (Atari) and utilize different actor-critic learning schemes, which cannot be directly compared with our work on continuous control.
>
> [3] Chen et al. TransDreamer: Reinforcement learning with transformer world models. 2022.
>
> **Q4**: Clarification on sentences
>
> - Line 234: Our 'IPV w/ vanilla WM' baseline mainly has two major differences with DreamerV2: in-the-wild video pre-training and video-based intrinsic bonus. Although 'IPV w/ vanilla WM' outperforms DreamerV2, our ablation study shows that this baseline can only benefit from the intrinsic bonus, but not in-the-wild pre-training.
> - Line 182: As shown, the intrinsic bonus is essential for learning efficiency, but it is computed using an ever-changing replay buffer during training. An additional predictor for pure task reward can force the dynamics model to encode task-relevant information, regardless of the intrinsic reward drifts, which helps representation learning. We do not downweight intrinsic reward since it needs extra hyperparameter tuning and, more importantly, may hurt exploration and learning efficiency.
>
> **Q5**: Minor questions
>
> - KL term: When t=1, the terms should be $\text{KL}[q(z_1|o_1)\|p(\hat{z}_1)]$ and $\text{KL}[q(s_1|z_1)\|p(\hat{s}_1)]$. Following the implementation of Dreamer, priors and posteriors of $z_1$ and $s_1$ are predicted with dummy previous states $z_0, s_0$ (all-zero initial states of RSSM) and actions $a_0$ (all-zero too), which unifies the implementations for t=1 and t>1.
> - Legends and Fig. 3: We appreciate the suggestions and will use check marks for the legend and revise Fig. 3 to make it clearer in a future revision.
>
> **Q6**: Limitations
>
> We apologize for the limited discussion of limitations. We will add a detailed discussion in a future revision. Please see $\underline{\text{Q1 in the global response}}$ for the revised discussion on limitations.

---

> ### Comment · Area_Chair_NT8x · 2023-08-17
> **Your response to the author rebuttal is missing. Please do it ASAP.**
>
> Dear Reviewer,
>
> The author has posted their rebuttal, but you have not yet posted your response. Please post your thoughts after reading the rebuttal and other reviews as soon as possible. All reviewers are requested to post this after-rebuttal-response.

---

> ### Author Response · Authors · 2023-08-19
> **Request of Reviewer's attention and feedback**
>
> Dear Reviewer JHvL,
>
> Thanks again for your dedication to reviewing our paper.
>
> We write to kindly remind you that this is the last few days of the Reviewer-author discussion period. We have made every effort to address the concerns you suggested and improve our paper:
>
> - We **clarify our usage of the term 'context' to dispel misunderstandings**. 'Contexts' in our paper stands only for spatial contexts, whereas the temporal context is handled by standard RSSMs. We provide additional results in the global response to show that our model can clearly separate contexts and dynamics.
> - We **experiment with other context frame selection methods** and show that our method is adequate for context modeling in our experimental benchmarks.
> - We **explain why transformer-based architecture is orthogonal to our technical contributions**. Nevertheless, we provide additional comparison results against transformer-based methods, where our method still performs the best.
> - We clarify and revise our writing on several potentially misleading sentences.
>
> Please kindly let us know if you have any remaining questions. If our responses have addressed your concerns, would you please consider re-evaluating our work based on the updated information? Looking forward to your reply.
>
> Sincerely,
>
> Authors

---

> > ### Comment · Reviewer_JHvL · 2023-08-21
> >
> > Thank you for providing a detailed response. I have read the answer and other reviewers' comments. To a good extent, the rebuttal resolves my concerns, and I am happy to increase my score. However, I still think the comparison between Transformer-based architecture should be investigated deeper, as, from the definition of the paper, all transformer-based architecture could be regarded as ``contextualized``.  Besides, the transformer-based architecture might be easier to incorporate pre-training on video without any mechanism to add layers (just changing conditioning variables might be enough).

---

> > > ### Author Response · Authors · 2023-08-21
> > > **Appreciation for Your Support and Constructive Feedback**
> > >
> > > Dear Reviewer JHvL,
> > >
> > > We sincerely appreciate your careful review of our rebuttal and your thoughtful reconsideration of your assessment. Your feedback has been invaluable in strengthening our paper. We acknowledge that while our contextualized image decoder's cross-attention mechanisms resemble 'transformer layers' for contextual information conditioning, your suggestion to thoughtfully craft a dedicated transformer-based architecture is essential. Furthermore, your perspective on the flexibility of transformer-based architecture to incorporate pre-training on video by changing conditioning variables is also truly insightful. As previously discussed, the combination of our pre-training framework with a transformer architecture holds great potential, and we will certainly add discussion regarding this in our revised paper, and delve deeper into this aspect in future work.
> > >
> > > Thank you for your constructive input.
> > >
> > > Best regards,
> > >
> > > Authors

---

### Official Review · Reviewer_kJR6 · 2023-07-10

**Soundness:** 4 excellent
**Presentation:** 3 good
**Contribution:** 4 excellent
**Rating:** 8
**Confidence:** 5

**Summary:**

This submission presents the contextualized world model (ContextWM), a framework for leveraging in-the-wild videos for pre-training of a world model to be used in model-based reinforcement learning. Following the work from Seo et al. (2022), the authors pre-train an action-free version of the recurrent state-space model (RSSM) with two important modifications: 1) a context encoder processes a randomly sampled frame of the input video to provide context features, which the decoder can directly access to better reconstruct static visual details, enabling the dynamics model, on which the decoder is also conditioned, to focus on temporally varying information. 2) the authors also opt for using a dual-reward predictor during fine-tuning, which predicts the pure task reward in addition to the combined task and video-based intrinsic novelty reward proposed by Seo et al. (2022). This facilitates task-relevant representation learning.
The ContextWM is evaluated on Meta-world, the remastered DeepMind Control Suite, and a task with varying weather conditions in the CARLA driving simulator. In most benchmarks, ContextWM shows significant improvements in terms of sample efficiency or final performance. An ablation study and other analytical experiments further show the effectiveness of ContextWM and its design decisions.

**Strengths:**

The proposed contextualized world model is novel and the design decisions are well motivated.
For the most part the description of the method and experimental setup is very clear. The performance improvements are significant and in some cases very impressive. The qualitative analysis provides some interesting insights, for instance the clear separation of video representations for two videos with contrastive labels in Figure 8b.

**Weaknesses:**

1. As the main contribution the cross-attention to the context should be explained in a bit more detail. Since the authors mention U-Nets, I wonder whether the decoder attends to context features at each of the corresponding three stages shown in Table 1 in the supplementary material or only at the decoder input level?

**Questions:**

2. Why did you choose BatchNorm instead of LayerNorm? Did you experiment with both?
1. maybe use a check mark in the legend of Figure 6b instead of the circle to indicate "with pretraining".
1. It'd be very interesting to see more examples with other contrastive labels for the Video representations experiment in Section 5.5.

## Acknowledgement of rebuttal
I have read the rebuttal, other reviews. My relatively minor concerns have been addressed, given that the authors have provided some requested clarifications, a discussion of limitations, and additional insightful experiments. I strongly believe this paper should be accepted.

**Limitations:**

The discussion of limitations is very *limited*. The discussion section mostly mentions future work on scaling and exploring other pre-training objectives.

---

> ### Author Rebuttal · Authors · 2023-08-10
>
> We sincerely thank Reviewer kJR6 for providing a detailed review, valuable suggestions, and a positive evaluation of our paper.
>
> **Q1**: Details of cross-attentions in the decoder
>
> In $\underline{\text{Appendix C.3 in the supplementary material}}$, we have elaborated the details of how our multi-scale context features are connected to the decoder in a U-Net style:
>
> > The outputs of the last residual block of two stages in the context encoder (stage2 and stage3) before average pooling (thus in the shape of 16 × 16 and 8 × 8, respectively) are passed to the corresponding residual block of the image decoder and used to augment the incoming decoder features with cross-attention.
>
> We do not use the 32 × 32 features from stage 1 due to the quadratic memory complexity of cross-attention. We will clarify these details in the main text in a future revision.
>
> **Q2**: BatchNorm vs LayerNorm
>
> We chose BatchNorm as it is the dominant normalization technique in CNNs. Note that our technical contributions are orthogonal to the choice of visual backbones. Exploring transformer-based backbones (e.g., ViTs), which are usually equipped with LayerNorm, is left for future work. We have experimented LayerNorm replacing BatchNorm in our architecture. Results in $\underline{\text{Fig. 5 of the global response attachment}}$ indicate that our architecture is robust to the choices.
>
> **Q3**: Additional visualization of video representations
>
> We have provided additional examples in $\underline{\text{Fig. 2 of the global response attachment}}$. Given videos with two distinct labels, '*moving away from something with your camera*' and '*approaching something with your camera*', our ContextWM provides a clear separation of video representations while vanilla WM fails.
>
> **Q4**:  Suggestion on the legends
>
> We appreciate this suggestion and will use a checkmark instead of the O mark to indicate "with pre-training" in a future revision.
>
> **Q5**: Limitations
>
> We apologize for the limited discussion of limitations. We will add a detailed discussion in a future revision. Please see $\underline{\text{Q1 in the global response}}$ for the revised discussion on limitations.

---

> > ### Comment · Reviewer_kJR6 · 2023-08-13
> >
> > I thank the authors for the clarifications and conducting additional insightful experiments (e.g. the model roll-outs with changed context features). I believe the paper should be accepted and I encourage my fellow reviewers to carefully study the author responses, since some of the criticism was based on misunderstanding, which the authors tried to resolve with detailed explanations.

---

> > > ### Author Response · Authors · 2023-08-13
> > > **Appreciation for Your Support**
> > >
> > > Dear Reviewer kJR6,
> > >
> > > Thank you sincerely for your positive and encouraging feedback. We greatly appreciate your recognition of our efforts to address concerns and provide clarifications and we hope our explanations will help dispel any misunderstandings. Your recommendation for acceptance boosts our confidence in the value of our work.
> > >
> > > Best regards,
> > >
> > > Authors

---

### Author Rebuttal · Authors · 2023-08-10

## Global Response to All Reviewers

We would like to thank the reviewers for their detailed comments. This paper aims to pre-train a broadly generalizable world model from in-the-wild videos to boost sample-efficient learning of downstream visual control tasks. Extensive experiments on large-scale video datasets and various visual control domains have demonstrated the effectiveness of our proposed In-the-wild Pre-training from Videos (IPV) with ContextWM.

We have made every effort to address all the reviewers' concerns and responded to the individual reviews below. We have also answered common questions raised by the reviewers in this global response.

Note that **all the new figures supplementary to all responses are included in the PDF attachment of this global response**. We only present results on part of the tasks due to limited time and computational resources.

**Q1**: Revise the discussion on limitations

We apologize for the limited discussion of limitations. We will add the following expanded discussion on limitations in a future revision:

> **Limitations and future work.** One limitation of our current method is that a randomly selected single context frame may not be sufficient to capture complete contextual information of scenes in the real world. Consequently, selecting and incorporating multiple context frames as well as multimodal information [47] for better context modeling need further investigation. Our work is also limited by medium-scale sizes in terms of both world models and pre-training data, which may hinder learning broadly applicable knowledge. Given that, an important direction is to systematically examine the scalability of our method by leveraging scalable architectures like Transformers [36, 54] and massive-scale video datasets [12, 37]. Lastly, our work focuses on pre-training world models via generative objectives, which use massive parameters inefficiently on image reconstruction to model intricate contexts. Exploring alternative pre-training objectives, such as contrastive learning [40, 7] or self-prediction [52], could further release the potential of IPV by eliminating heavy components on context modeling and focusing on dynamics modeling.

**Q2**: Particularly significant performance gains on DMCR tasks

Our method obtains considerable performance gains on DMCR tasks. The main reason is that DMCR is a purposefully designed benchmark, which measures visual generalization and requires the agent to extract task-relevant information as well as ignore visual distractors. Our ContextWM has the advantage of separately modeling contexts (task-irrelevant in DMCR) and dynamics (task-relevant in DMCR), which avoids wasting the capacity of dynamics models in modeling low-level visual details. Furthermore, pre-training with in-the-wild videos enables our models to eliminate diverse distractors and capture shared motions, which is essential for visual generalization in RL. In contrast, vanilla WM needs to model complicated contexts and dynamics in an entangled manner,  which adds difficulty to dynamics learning and behavior learning on these features.

**Qualitative analysis**: To demonstrate the ability of ContextWM to separate context and dynamics modeling, we provide additional video prediction results in **$\underline{\text{Fig. 1 of the global response attachment}}$**. In the _Context Shift_ row of the figure, we sample a random frame from another trajectory to replace the original context and leave dynamics the same as the _IPV w/ ContextWM_ row to conduct a **compositional decoding analysis**. We can see that after shifting the context, ContextWM correctly combines contextual information from the new context with dynamics information from the original trajectory. These results show that our model finetuned on the DMCR domain has successfully learned disentangled representations of contexts and dynamics. The vanilla WM, on the other hand, suffers from learning entangled features and, as a result, makes poor predictions about the environment transitions.

**General-purpose framework**: We also emphasize that, motivated by separating contexts and enhancing temporal dynamics modeling, our proposed IPV w/ ContextWM is a general-purpose framework and, as shown, can obtain adequate performance gain on various benchmarks that have more complicated entangling of contexts and dynamics beyond DMCR.

**Q3**: Suggestion on the legends and Fig. 3 of main paper

Thanks for the valuable suggestions. We will use check marks for the legends to indicate 'with pre-training' and revise Fig. 3 to make the architecture clearer in a future revision.

---

### Decision · Program_Chairs · 2023-09-21

**Decision:**

Accept (poster)

**Comment:**

The paper investigates the pre-training of world models using in-the-wild videos to enhance the sample efficiency in model-based reinforcement learning (MBRL) for tasks such as robotic manipulation and autonomous driving. To tackle the complexities found in these videos, it introduces Contextualized World Models (ContextWM), which explicitly model both the context and dynamics to facilitate knowledge transfer across distinct scenes.

Overall, reviewers have praised the paper for its clarity, methodological soundness, and compelling experimental results. The novel idea of contextual encoding is well-received, and the experiments showcase significant performance improvements in multiple domains. Some reviewers had concerns regarding technical details and comparisons to prior work, but the rebuttal has addressed these issues to some extent. There are still some points that could be clarified further, such as the role of cross-attention in context modeling and a deeper discussion with transformer-based world models. Yet, the consensus leans towards acceptance, particularly given the additional clarifications and experiments presented in the rebuttal. Therefore, the paper is recommended for acceptance with the understanding that the authors will continue to refine it by addressing the remaining weaknesses.